# NEURAL SIMULATED ANNEALING

## ABSTRACT

Simulated annealing (SA) is a stochastic global optimisation technique applicable to a wide range of discrete and continuous variable problems. Despite its simplicity, the development of an effective SA optimiser for a given problem hinges on a handful of carefully handpicked components; namely, neighbour proposal distribution and temperature annealing schedule. In this work, we view SA from a reinforcement learning perspective and frame the proposal distribution as a policy, which can be optimised for higher solution quality given a fixed computational budget. We demonstrate that this *Neural SA* with such a learnt proposal distribution outperforms SA baselines with hand-selected parameters on a number of problems: Rosenbrock's function, the Knapsack problem, the Bin Packing problem, and the Travelling Salesperson problem. We also show that Neural SA scales well to large problems while again outperforming popular off-the-shelf solvers in terms of solution quality and wall clock time.

## 1 INTRODUCTION

There are many different kinds of combinatorial optimisation (CO) problem, spanning bin packing, routing, assignment, scheduling, constraint satisfaction, and more. Solving these problems while sidestepping their inherent computational intractability has great importance and impact for the real world, where poor bin packing or routing lead to wasted profit or excess greenhouse emissions (Salimifard et al., 2012). General solving frameworks or *metaheuristics* for all these problems are desirable, due to their conceptual simplicity and ease-of-deployment, but require manual tailoring to each individual problem. One such metaheuristic is *Simulated Annealing* (SA) (Kirkpatrick et al., 1987), a simple, and equally very popular, iterative global optimisation technique for numerically approximating the global minimum of both continuous- and discrete-variable problems. While SA has wide applicability, this is also its Achilles' Heel, leaving many design choices to the user. Namely, a user has to design 1) neighbourhood proposal distributions, which define the space of possible transitions from a solution $\mathbf{x}_k$ at time $k$ to solutions $\mathbf{x}_{k+1}$ at time $k+1$, and 2) a temperature schedule, which determines the balance of exploration to exploitation. In this work, we mitigate the need for extensive finetuning of SA's parameters by designing a learnable proposal distribution speeding up convergence while limiting computational overhead to $\mathcal{O}(N)$ per step for problem size $N$.

CO algorithms fall into two categories: exact and approximate. Exact methods such as dynamic programming (Bellman, 1952) and branch-and-bound (Land & Doig, 1960) find global minimisers $\mathbf{x}^*$, but are slow due to the NP-hardness of many CO problems. Approximate algorithms therefore seek a good enough solution under practical computational constraints. SA is one such approximate algorithm. In recent years there has been an explosion of works (Bengio et al., 2018) in machine learning for CO (ML4CO). In ML4CO, a lot of work has focused on end-to-end neural architectures (Bello et al., 2016; Vinyals et al., 2017; Dai et al., 2017; Kool et al., 2018; Emami & Ranka, 2018; Bresson & Laurent, 2021). These work by brute force learning the instance to solution mapping—in CO these are sometimes referred to as *construction heuristics*. Other works focus on learning good parameters for classical algorithms, whether they be parameters of the original algorithm (Kruber et al., 2017; Bonami et al., 2018) or extra neural parameters introduced into the computational graph of classical algorithms (Gasse et al., 2019; Gupta et al., 2020; Kool et al., 2021; de O. da Costa et al., 2020; Wu et al., 2019b; Chen & Tian, 2019; Fu et al., 2021). Our method, *neural simulated annealing* (Neural SA) can be viewed as sitting firmly within this last category.

SA is an improvement heuristic; it navigates the search space of feasible solutions by iteratively applying (small) perturbations to previously found solutions. Figure 1 illustrates this for the Travelling

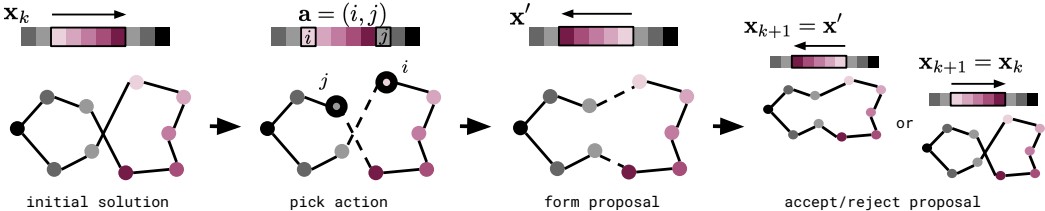

Figure 1: Neural SA pipeline for the TSP. Starting with a solution (tour) $\mathbf{x}_k$, we sample an action $\mathbf{a} = (i, j)$ from our learnable policy/proposal distribution, defining start $i$ and end $j$ points of a 2-opt move (replacing two old with two new edges). Each pane shows both the linear and graph-based representations for a tour. From $\mathbf{x}_k$ and $\mathbf{a}$ we form a proposal $\mathbf{x}'$ which is either accepted or rejected according to the traditional MH step used in SA. Accepted moves assign $\mathbf{x}_{k+1} = \mathbf{x}'$; whereas, rejected moves assign $\mathbf{x}_{k+1} = \mathbf{x}_k$.

Salesperson Problem (TSP), perhaps the most classic of NP-hard problems. In this work, we pose this as a Reinforcement Learning (RL) agent navigating an environment, searching for better solutions. In this light the proposal distribution is an optimisable quantity. Conveniently, our method inherits convergence guarantees from SA. We are able to directly optimise the proposal distribution using policy optimisation for both faster convergence and better solution quality under a fixed computation budget. We demonstrate Neural SA on four tasks: Rosenbrock's function, a toy 2D optimisation problem, where we can easily visualise and analyse what is being learnt; the Knapsack and Bin Packing problems, which are classical NP-hard resource allocation problems; and the TSP.

Our contributions are:

- We pose simulated annealing as a Markov decision process, bringing it into the realm of reinforcement learning. This allows us to optimise the proposal distribution in a principled manner. Our method inherits all the convergence guarantees of vanilla simulated annealing.
- We demonstrate superior performance to off-the-shelf CO tools on the Knapsack, Bin Packing, and Travelling Salesperson problems, in terms of solution quality and wall-clock time.
- We show our methods transfer to problems of different sizes, and also perform well on problems up to $40\times$ larger than the ones used for training.
- Our method is competitive within the ML4CO space, only using a very lightweight neural architecture, with number of learnable parameters of the order of 100s or fewer.

## 2 BACKGROUND AND RELATED WORK

Here we outline the basic simulated annealing algorithm and its main components. Then we provide an overview of prior works in the machine learning literature which have sought to learn parts of the algorithm or where SA has found uses in machine learning.

**Combinatorial optimisation** A combinatorial optimisation problem is defined by a triple $(\mathbf{\Psi}, \mathcal{X}, E)$ where $\psi \in \mathbf{\Psi}$ are problem instances (city locations in the TSP), $\mathcal{X}$ is the set of feasible solutions given $\psi$ (Hamiltonian cycles in the TSP) and $E : \mathcal{X} \times \mathbf{\Psi} \to \mathbb{R}$ is an *energy function* (tour length in the TSP). Without loss of generality, the task is to minimise the energy $\min_{\mathbf{x} \in \mathcal{X}} E(\mathbf{x}; \psi)$. CO problems are in general NP-hard, meaning that there is no known algorithm to solve them in time polynomial in the number of bits that represents a problem instance.

**Simulated Annealing** Simulated annealing (Kirkpatrick et al., 1987) is a metaheuristic for CO problems. It builds an inhomogeneous Markov chain $\mathbf{x}_0 \to \mathbf{x}_1 \to \mathbf{x}_2 \to \cdots$ for $\mathbf{x}_k \in \mathcal{X}$, asymptotically converging to a minimizer of $E$. The stochastic transitions $\mathbf{x}_k \to \mathbf{x}_{k+1}$ depend on two quantities: 1) a proposal distribution, and 2) a temperature schedule. The proposal distribution $\pi : \mathcal{X} \to \mathbb{P}(\mathcal{X})$, for $\mathbb{P}(\mathcal{X})$ the space of probability distributions on $\mathcal{X}$, suggests new states in the chain. It perturbs current solutions to new ones, potentially leading to lower energies immediately or later on. After perturbing a solution $\mathbf{x}_k \to \mathbf{x}'$, a Metropolis–Hastings (MH) step (Metropolis et al., 1953; Hastings, 1970) is executed. This either accepts the perturbation ($\mathbf{x}_{k+1} = \mathbf{x}'$) or

rejects it ($\mathbf{x}_{k+1} = \mathbf{x}_k$)—see Algorithm 1 for details. The target distribution of the MH step has form $p(\mathbf{x}|T_k) \propto \exp\{-E(\mathbf{x})/T_k\}$, where $T_k$ is the *temperature* at time $k$. In the limit $T_k \to 0$, this distribution tends to a sum of Dirac deltas on the minimisers of the energy. The temperature is annealed, according to the temperature schedule, $T_1, T_2, ...$, from high to low, to steer the target distribution smoothly from broad to peaked around the global optima. The algorithm is outlined in Algorithm 1. Under certain regularity conditions and provided the chain is long enough, it will visit the minimisers almost surely (Geman & Geman, 1984). More concretely,

$$\lim_{k \to \infty} P\left(\mathbf{x}_k \in \arg\min_{\mathbf{x} \in \mathcal{X}} E(\mathbf{x}; \boldsymbol{\psi})\right) = 1. \tag{1}$$

Despite this guarantee, practical convergence speed is determined by $\pi$ and the temperature schedule, which are hard to fine-tune. There exist problem-specific heuristics for setting these (Pereira & Fernandes, 2004; Cicirello, 2007), but in this paper we propose to learn the proposal distribution.

## 2.1 SIMULATED ANNEALING AND MACHINE LEARNING

A natural way to combine machine learning and simulated annealing is to design local improvement heuristics that feed off each other. Notable examples are Cai et al. (2019) and Vashisht et al. (2020) where RL is used to find good initial solutions later refined by standard SA. The two methods only interact via shared solutions, fundamentally different to our approach, as we focus on augmenting SA with RL optimisable components. In fact, our method is perfectly compatible with theirs and any other SA application. Other works proposed the optimising components of SA. In particular, Blum et al. (2020) study the design of optimal cooling schedules using data, deriving theoretical guarantees on sample complexity. However, the authors do not provide empirical evaluations and focus exclusively on the cooling schedule while we concentrate on the proposal distribution. Similarly, Wauters et al. (2020) and Khairy et al. (2020) use RL to find optimal parameters for quantum variants of SA but their methods are not readily applicable to standard SA.

More closely to our method, other approaches improve the proposal distribution. In Adaptive Simulated Annealing (ASA) (Ingber, 1996) the proposal distribution is not fixed but evolves throughout annealing process as a function of the variance of observed solution fitness. ASA improves the convergence of standard SA but is not learnable like Neural SA. To the best of our knowledge, Marcos Alvarez et al. (2012) are the only others to learn the proposal distribution for SA, but they rely on supervised learning, requiring high quality solutions or good search strategies to imitate, both expensive to compute. In contrast, Neural SA is fully unsupervised, thus easier to train and extend to different CO tasks. Finally, many Monte Carlo methods use a a proposal distribution. Noé et al. (2019) and Albergo et al. (2019) recently studied how to learn a proposal distribution of an MCMC chain for sampling the Boltzmann distribution of a physical system. While their results serve as motivation for our methods, we investigate a completely different context and set of applications.

Our work falls under bi-level optimisation methods, where an outer optimisation loop finds the best parameters of an inner optimisation. This encompasses situations such as learning the parameters (Rere et al., 2015) or hyperparameters of a neural network optimiser (Maclaurin et al., 2015; Andrychowicz et al., 2016) and meta-learning (Finn et al., 2017). However, most recent approaches assume differentiable losses on continuous state spaces Likhosherstov et al. (2021); Ji et al. (2021); Vicol et al. (2021), while we focus on the more challenging CO setting. We note, however, methods in Vicol et al. (2021) are based on evolution strategies and could be used in the discrete setting.

## 2.2 MARKOV DECISION PROCESSES

Simulated annealing naturally fits into the Markov Decision Process (MDP) framework as we explain below. An MDP $\mathcal{M} = (\mathcal{S}, \mathcal{A}, R, P, \gamma)$ consists of *states* $s \in \mathcal{S}$, *actions* $a \in \mathcal{A}$, an *immediate reward function* $R : \mathcal{S} \times \mathcal{A} \times \mathcal{S} \to \mathbb{R}$, a *transition kernel* $P : \mathcal{S} \times \mathcal{A} \to \mathbb{P}(\mathcal{S})$, and a *discount factor* $\gamma \in [0, 1]$. On top of this MDP we add a stochastic *policy* $\pi : \mathcal{S} \to \mathbb{P}(A)$. The policy and transition kernel together define a length-$K$ trajectory $\tau = (s_0, a_0, s_1, a_1, ..., s_K)$, which is a sample from the distribution $P(\tau|\pi) = \rho_0(s_0) \prod_{k=0}^{K-1} P(s_{k+1}|s_k, a_k)\pi(a_k|s_k)$ and where $s_0 \sim \rho_0$ is sampled from the *start-state distribution* $\rho_0$. One can then define the *discounted return* $R(\tau) = \sum_{k=0}^{K-1} \gamma^t r_k$ over a trajectory, where $r_k = R(s_k, a_k, s_{k+1})$. We say that we have solved an MDP if we have found a policy that maximises the expected return $\mathbb{E}_{\tau \sim P(\tau|\pi)}[R(\tau)]$.

---

**Algorithm 1** Neural simulated annealing. To get back to vanilla SA, replace the parametrised proposal distribution $\pi_\theta$ with a uniform distribution $\pi$ over neighbourhoods $\mathcal{N}(\bullet)$.

---

**Require:** Initial state $\mathbf{s}_0 = (\mathbf{x}_0, \boldsymbol{\psi}, T_0)$, proposal distribution $\pi$, transition function $P$, temperature schedule $T_1 \geq T_2 \geq T_3 \geq ...$, energy function $E(\bullet; \boldsymbol{\psi})$
   **for** $k = 1 : K$ **do**
      $\mathbf{a} \sim \pi_\theta(\mathbf{s}_k)$                                                     $\triangleright$ Sample action
      $u \sim \text{Uniform}(u; 0, 1)$                                $\triangleright$ Metropolis–Hastings step
      **if** $u < \exp\left\{-(E(\mathbf{x}'; \boldsymbol{\psi}) - E(\mathbf{x}_k; \boldsymbol{\psi}))/T_k\right\}$ **then**
         $\mathbf{s}_{k+1} \leftarrow (\mathbf{x}', \boldsymbol{\psi}, T_{k+1})$                                      $\triangleright$ Accept
      **else**
         $\mathbf{s}_{k+1} \leftarrow (\mathbf{x}_k, \boldsymbol{\psi}, T_{k+1})$                                     $\triangleright$ Reject
      **end if**
   **end for**

---

## 3 METHOD

Here we outline our approach to learn the proposal distribution. First we define an MDP corresponding to SA. We then show how the proposal distribution can be optimised and provide a justification that this does not affect convergence guarantees of the classical algorithm.

### 3.1 MDP FORMULATION

We formalise SA as an MDP, defining states $\mathbf{s} = (\mathbf{x}, \boldsymbol{\psi}, T) \in \mathcal{S}$ for $\boldsymbol{\psi}$ a parametric description of the problem instance as in Section 2, and $T$ the instantaneous temperature. Examples are in Section 4. Our actions $\mathbf{a} \in \mathcal{A}$ perturb $(\mathbf{x}, \boldsymbol{\psi}, T) \mapsto (\mathbf{x}', \boldsymbol{\psi}, T)$, where $\mathbf{x}' \in \mathcal{N}(\mathbf{x})$ is a solution in the neighbourhood of $\mathbf{x}$. It is common to define small neighbourhoods, to limit energy variation from one state to the next. This heuristic discards exceptionally good and exceptionally bad moves, but given that bad moves are commoner than good ones, it generally leads to faster convergence.

The MH step in SA can be viewed as a stochastic transition kernel, governed by the current temperature of the system, with transition probabilities following a Gibbs distribution and dynamics

$$\mathbf{x}_{k+1} = \begin{cases} \mathbf{x}', & \text{with probability } p \\ \mathbf{x}_k, & \text{with probability } 1 - p, \end{cases} \qquad \text{where } p = \min\left\{1, e^{-\frac{1}{T_k}(E(\mathbf{x}'; \boldsymbol{\psi}) - E(\mathbf{x}_k; \boldsymbol{\psi}))}\right\}. \quad (2)$$

This defines a transition kernel $P(\mathbf{s}_{k+1}|\mathbf{s}_k, \mathbf{a})$, where $\mathbf{s}_{k+1} = (\mathbf{x}', \boldsymbol{\psi}, T)$ or $\mathbf{s}_{k+1} = (\mathbf{x}_k, \boldsymbol{\psi}, T)$, the two outcomes of the MH step. For rewards, we use either the immediate gain $E(\mathbf{x}_k; \boldsymbol{\psi}) - E(\mathbf{x}_{k+1}; \boldsymbol{\psi})$ or the primal reward $-\delta_{k=K-1} \min_{\mathbf{x} \in \mathbf{x}_{1:k}} E(\mathbf{x}; \boldsymbol{\psi})$. We explored training with two different methods: Proximal Policy Optimisation (PPO) (Schulman et al., 2017) and Evolution Strategies (ES) Salimans et al. (2017). The immediate gain works best with PPO, where at each iteration of the rollout, the immediate gain gives fine-grained feedback on whether the previous action helped or not. The primal reward works best with ES, because it is non-local, returning the minimum along an entire rollout $\tau$ at the very end. We explored using the acceptance count but found that this sometimes led to pathological behaviours. We also tried the primal integral (Berthold, 2013), which encourages finding a good solution fast, but found we could not get training dynamics to converge.

### 3.2 POLICY NETWORK ARCHITECTURE

SA chains are long. It is because of this that we need as lightweight a policy architecture as possible. Furthermore, this architecture should have the capacity to scale to varying numbers of inputs, so that we can transfer experience across problems of different size $N$. We opt for a very simple network, shown in Figure 2. For each dimension of the problem we map the state $(\mathbf{x}, \boldsymbol{\psi}, T)$ into a set of features. For all problems we try, there is a natural way to do this. Each feature is fed into an MLP, embedding it into a logit space, followed by a softmax function to yield probabilities. The complexity of this architecture scales linearly with $N$, which is important since we plan to evaluate it many times. A notable property of this architecture is that is is permutation equivariant (Zaheer et al., 2017), an important requirement for CO problems. Note that our model is a permutation equivariant set-to-set mapping, but we have not used attention or other kinds of pairwise interaction to keep the computational complexity linear in the number of items.

**Convergence** Convergence of SA to the optimum in the infinite time limit requires the Markov chain of the proposal distribution to be irreducible (van Laarhoven & Aarts, 1987, Thm 6, Chap 3), meaning that for any temperature, any two states are reachable through a sequence of transitions with positive conditional probability under $\pi$. Our neural network policy satisfies this condition as long as the softmax layer does not assign zero probability to any state, a condition which is met in practice. Thus Neural SA inherits convergence guarantees from SA.

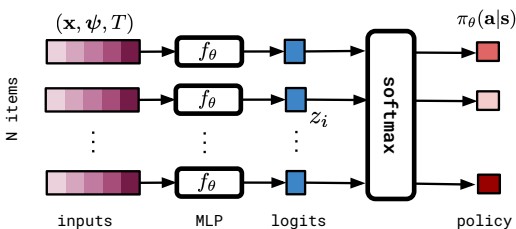

Figure 2: (a) Policy network: The same MLP is applied to all inputs pointwise. We use this network in all CO problems.

## 4 EXPERIMENTS

We evaluate our method on 4 tasks—Rosenbrock's function, the Knapsack, Bin Packing, and Travelling Salesperson problems—fixing the architecture and overall hyperparameters of Neural SA. This shows the wide applicability and ease of use of our method. For each task (except for Rosenbrock's function) we test Neural SA on problems of different size $N$, training only on the smallest. Similarly, we consider rollouts of different lengths, training only on short ones. This accelerates training, showing Neural SA's generalisation capabilities. This type of transfer learning is rare in ML4CO, and is a merit of our lightweight, equivariant architecture. In all experiments, we adopt an exponential multiplicative cooling schedule as originally proposed by Kirkpatrick et al. (1987), with $T_k = \alpha^k T_0$. $\alpha$ is computed for set $T_0$ and $T_K$. This allows us to vary the rollout length while maintaining the same range of temperatures for every run. Exact training details are in the Appendix.

### 4.1 THE ROSENBROCK FUNCTION

The Rosenbrock function is non-convex over Euclidean space. Of course, gradient-based optimisers are more suited to this problem, but we use it as toy example to showcase the properties of Neural SA. Our policy is an axis-aligned Gaussian $\pi_\theta(\mathbf{a}|\mathbf{s}) = \mathcal{N}(\mathbf{a}; \mathbf{0}, \sigma_\theta^2(\mathbf{s}_i))$, with MLP-parameterised variance $\sigma_\theta^2$ of shape $2 \rightarrow 16 \rightarrow 2$ and a ReLU in the middle. Proposals are $\mathbf{x}' = \mathbf{x} + \mathbf{a}$, and the state is $\mathbf{s}_k = (\mathbf{x}_k, a, b, T_i)$. An example rollout is in Figure 3a. Mathematically the function is

$$E(x_0, x_1; a, b) = (a - x_0)^2 + b(x_1 - x_0^2)^2, \tag{3}$$

which has global minimum at $\mathbf{x} = (a, a^2)$. We contrast Neural SA against vanilla SA with fixed proposal distribution, i.e. $\sigma(\mathbf{s}_i) = \sigma$, for different $\sigma$ averaged over $2^{17}$ problem instances. This shows in Figure 3d that no constant variance policy can outperform an adaptive policy on this problem. Plots of acceptance ratio in Figure 3b show Neural SA has higher acceptance probability early in the rollout, a trend we observed in all experiments, suggesting its proposals are skewed towards lower energy solutions than standard SA. Figure 3c shows the variance network $\sigma_\theta^2(\mathbf{s}_i)$ as a function of time. It has learnt to make large steps until hitting the basin, whereupon large moves will be rejected with high probability, so variance must be reduced.

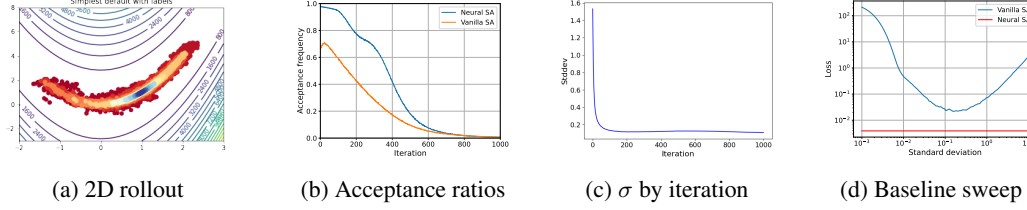

| (a) 2D rollout | (b) Acceptance ratios | (c) $\sigma$ by iteration | (d) Baseline sweep |

Figure 3: Results on Rosenbrock's function: (a) Example trajectory, moving from red to blue, showing convergence around the minimiser at (1,1) (b) Neural SA has higher acceptance ratio than the baseline, a trend observed in all experiments, (c) Standard deviation of the learned policy as a function of iteration. Large initial steps offer great gains followed by small exploitative steps, (d) A non-adaptive vanilla SA baseline cannot match an adaptive one, no matter the standard deviation.

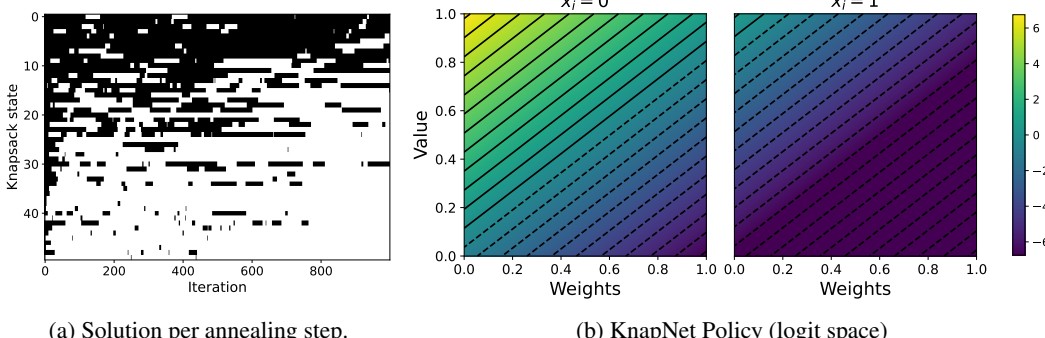

(a) Solution per annealing step.       (b) KnapNet Policy (logit space)

Figure 4: (a) Exemplar knapsack rollout on KNAP50: Each row is an item and each column is a subsequent iteration. Items absent from the knapsack are in black and those present are in white. Items are ordered by value-to-weight ratio in ascending order top to bottom. We see that light, valuable objects are held in the knapsack longer than heavy, valueless ones. (b) Knapsack logits: Policy logits for $x_i = 0$ and $x_i = 1$ are shown in each pane. Light valuable objects are favoured to insert. Once inserted the policy downweights an object's probably of flipping state again. Interestingly, the ejection probability of heavy, valueless objects is low, perhaps because this only makes sense close to overflowing, but the policy does not receive free capacity as a feature.

## 4.2 KNAPSACK PROBLEM

The Knapsack problem is a classic CO problem in resource allocation. Given a set of $N$ items, each of a different value $v_i > 0$ and weight $w_i > 0$, the goal is to find a subset that maximises the sum of values while respecting a limit $W$ on its total weight. This has corresponding integer linear program

$$\text{minimise } E(\mathbf{x}; \boldsymbol{\psi}) = -\sum_{i=0}^{N-1} v_i x_i, \qquad \text{subject to } \sum_{i=0}^{N-1} w_i x_i \leq W, \qquad x_i \in \{0, 1\}. \qquad (4)$$

Solutions are represented as a binary vector $\mathbf{x}$, with $x_i = 0$ for 'out of the bin' and $x_i = 1$ for 'in the bin'. Our proposal distribution flips individual bits, one at a time, with the constraint that we cannot flip $0 \mapsto 1$ if the bin capacity will be exceeded. The neighbourhood of $\mathbf{x}_k$ is thus all feasible solutions at a Hamming distance of 1 from $\mathbf{x}_k$. We use the proposal distribution described in Section 3.2 and illustrated in Figure 2, consisting of a pointwise embedding of each item—its weight, value, occupancy bit, the knapsack's overall capacity, and global temperature—into a logit-space, followed by a softmax. Mathematically the policy and state–action to proposal mapping are

$$\pi_\theta(i|\mathbf{s}) = \text{softmax}\,(\mathbf{z})_i\,, \qquad z_i = f_\theta([x_i, w_i, v_i, W, T]) \qquad (5)$$

$$\mathbf{x}' = \mathbf{x} + \text{onehot}(i) \mod 2. \qquad (6)$$

where $f_\theta$ is a very small two-layer neural network $5 \to 16 \to 1$ with a ReLU nonlinearity between the two layers. Actions are sampled from the categorical distribution induced by the softmax, and cast to one-hot vectors onehot($i$).

Neural networks have been used to solve the Knapsack Problem in Vinyals et al. (2017), Nomer et al. (2020), and Bello et al. (2016). We follow the setup of Bello et al. (2016), honing in on 3

Table 1: Average cost of solutions for the Knapsack Problem across five random seeds and, in parentheses, optimality gap to best solution found among solvers. Bigger is better. *Values as reported by Bello et al. (2016) for reference.

| | Random Search | Bello RL | Bello AS | SA | Ours (PPO) | Ours (ES) | Greedy | OR-Tools |
|---|---|---|---|---|---|---|---|---|
| KNAP50 | 17.91* | 19.86* | 20.07* | 18.90 (5.82%) | 19.87 (0.99%) | 19.95 (0.65%) | 19.94 (0.60%) | **20.12** (0.00%) |
| KNAP100 | 33.23* | 40.27* | 40.50* | 36.75 (9.26%) | 39.92 (1.43%) | 39.90 (1.48%) | 40.17 (0.81%) | **40.41** (0.00%) |
| KNAP200 | 35.95* | 57.10* | 57.45* | 48.88 (14.91%) | 55.65 (3.13%) | 55.58 (3.25%) | 57.30 (0.26%) | **57.65** (0.00%) |
| KNAP500 | - | - | - | 126.94 (11.93%) | 139.66 (3.10%) | 141.01 (2.17%) | 143.77 (0.25%) | **144.14** (0.00%) |
| KNAP1K | - | - | - | 254.44 (11.96%) | 280.39 (2.98%) | 282.46 (2.26%) | 288.64 (0.13%) | **289.01** (0.00%) |
| KNAP2K | - | - | - | 507.83 (12.03%) | 560.40 (2.92%) | 563.75 (2.34%) | 576.89 (0.06%) | **577.28** (0.00%) |

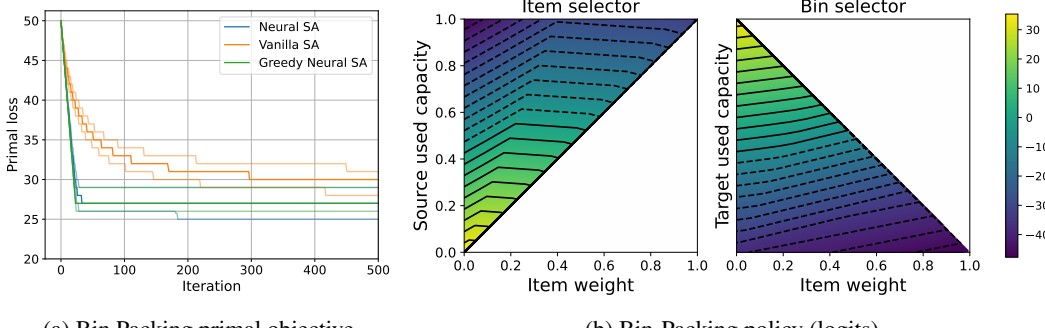

(a) Bin Packing primal objective                    (b) Bin-Packing policy (logits)

Figure 5: (a) Plot of the BIN50 primal objective for vanilla, Neural, and Greedy Neural SA. 25th, 50th, and 75th percentiles shown: Greedy and Neural SA have faster convergence than vanilla SA. Prolonged sampling improves over the greedy version. (b) Policy for the Bin Packing Problem: The bin packing policy consists of two networks, an item selector and a bin selector. The item selector uses item weight and bin used capacities to select an item to move. The bin selector then places this item in a bin, based on target bin fullness and selected item weight. The learnt policy is very sensible. The item selector looks for a light item in an under-full bin. The bin selector then place sthis in an almost-full bin. We mask bins with insufficient free capacity, hence the triangular logit-spaces.

self-generated datasets: KNAP50, KNAP100 and KNAP200. KNAP$N$ consists of $N$ items with weights and values generated uniformly at random in $(0, 1]$ and capacities $C_{50} = 12.5, C_{100} = 25$, and $C_{200} = 25$. We use OR-Tools (Perron & Furnon) to compute groundtruth solutions. Results in Table 1, show that Neural SA improves over vanilla SA by up to 10% optimality gap, and heuristic methods (Random Search) by much more. Neural SA falls slightly behind two methods by Bello et al. (2016), which use (1) a large attention-based pointer network with several orders of magnitude more parameters in Bello RL, and (2) this coupled with 5000 iterations of their Active Search method. It also falls behind a greedy heuristic for packing a knapsack based on the value-to-weight ratio. In Figure 4 we analyse the policy network and a typical rollout. It has learnt a mostly greedy policy to fill its knapsack with light, valuable objects, only ejecting them when full. This is line with the value-to-weight greedy heuristic. Despite not coming top among methods, we note Neural SA is typically within 1 - 3 % of the minimum energy, while not designed for this problem in particular.

### 4.3 BIN PACKING PROBLEM

The Bin Packing problem is similar to the Knapsack problem in nature. Here, one wants to pack *all* of $N$ items into the smallest number of bins possible, where each item $i \in \{1, \cdots, N\}$ has weight $w_i$, and we assume, without loss of generality, $N$ bins of equal capacity $W \geq \max_i(w_i)$—there would be no valid solution otherwise. If $x_{ij}$ denotes item $i$ occupying bin $j$, then the problem can be written as minimising an energy

$$\text{minimise } E(\mathbf{x}; \boldsymbol{\psi}) = \sum_{j=1}^{N} y_j, \tag{7}$$

$$\text{subject to } \underbrace{\sum_{i=0}^{N-1} w_i x_{ij} \leq W}_{\text{bin capacity constraint}}, \quad \underbrace{\sum_{j=0}^{N-1} x_{ij} = 1}_{\text{1 bin per item}}, \quad \underbrace{y_j = \min\left(1, \sum_{i=0}^{N-1} x_{ij},\right)}_{\text{bin occupancy indicator}}, \quad x_{ij} \in \{0, 1\} \tag{8}$$

where the constraints apply for all $i$ and $j$. We define the policy in two steps: we first pick an item $i$, and then select a bin $j$ to place it into. We can then write $\pi_{\theta,\phi}(\mathbf{a} = (i, j)|\mathbf{s}) = \pi_{\phi}(i|\mathbf{s})\pi_{\theta}(j|\mathbf{s}, i)$, which we parametrise as

$$\pi_{\theta}(i|\mathbf{s}) = \text{softmax}\left(\mathbf{z}^{\text{item}}\right)_i, \qquad z_i^{\text{item}} = f_{\theta}([w_i, c_{b(i)}, T]), \tag{9}$$

$$\pi_{\phi}(j|\mathbf{s}, i) = \text{softmax}\left(\mathbf{z}^{\text{bin}}\right)_j, \qquad z_j^{\text{bin}} = f_{\phi}([w_i, c_j, T]), \tag{10}$$

Table 2: Average cost of solutions for the Bin Packing Problem across five random seeds and, in parentheses, optimality gap to best solution found among solvers. Lower is better. We set a time out of 1 minute per problem for Or-Tools; * indicates only the trivial solution was found in this time.

|  | SA | Ours (PPO) | Ours (ES) | OR-Tools (SCIP) | FFD |
|---|---|---|---|---|---|
| BIN50 | 29.48 (10.37%) | 27.29 (2.17%) | 27.24 (1.98%) | **26.71** (0.00%) | 27.10 (1.46%) |
| BIN100 | 60.03 (13.46%) | 53.47 (1.06%) | 53.38 (1.29%) | 53.91 (1.89%) | **52.91** (0.00%) |
| BIN200 | 121.21 (16.27%) | 105.63 (1.32%) | 105.43 (1.13%) | 109.19 (4.74%) | **104.25** (0.00%) |
| BIN500 | 302.79 (17.81%) | 259.08 (0.80%) | 260.27 (1.26%) | 267.63 (4.13%) | **257.02** (0.00%) |
| BIN1000 | 605.09 (18.77%) | 512.66 (0.63%) | 516.84 (1.45%) | 1000* | **509.46** (0.00%) |
| BIN2000 | 1209.48 (18.82%) | **1017.88** (0.00%) | 1028.67 (1.06%) | 2000* | 1028.67 (1.06%) |

where $b(i)$ is the bin item $i$ is in before the action (in terms of $x_{ij}$, we have $x_{ib(i)} = 1$), $c_j$ is the free capacity of bin $j$ ($c_j = W - \sum_{i=1}^{N} w_i x_{ij}$), and both $f_\theta$ and $f_\phi$ are lightweight architectures $3 \rightarrow 16 \rightarrow 1$ with a ReLU nonlinearity between the two layers. We sample from the policy ancestrally, sampling first an item from $\pi_\theta(i|\mathbf{s})$, followed by a bin from $\pi_\phi(j|\mathbf{s}, i)$. Results in Table 2 show that our lightweight model is able to find a solution to about 1% higher energy than the minimum found by FFD Johnson (1973), a very strong heuristic for this problem (Rieck, 2021). We even see that we very often beat the SCIP (Gamrath et al., 2020a;b) optimizer in OR-Tools, which timed out on most problems. Figure 5a compared convergence speed of Neural SA with vanilla SA and a third option, Greedy Neural SA, which uses argmax samples from the policy. The learnt policy, visualised in Figure 5b has much faster convergence than the vanilla version. Again, we see that our method, although simple, is competitive with hand-design alternatives.

## 4.4 TRAVELLING SALESPERSON PROBLEM

Imagine you will make a round road-trip through $N$ cities and want to plan the shortest route visiting each city once; this is the Travelling Salesperson Problem (TSP) (Applegate et al., 2006). The TSP has been a long time favourite of computer scientists due to its easy description and NP-hardness. Here we use it as an example of a difficult CO problem. We compare with Concorde (Applegate et al., 2006) and LKH-3 (Helsgaun, 2000), two custom solvers for TSP. Given cities $i \in \{0, 1, ..., N-1\}$ with spatial coordinates $\mathbf{c}_i \in [0, 1]^2$, we wish to find a linear ordering of the cities, called a *tour*, denoted by the permutation vector $\mathbf{x} = (x_0, x_1, ..., x_{N-1})$ for $x_i \in \{0, 1, ..., N-1\}$ such that

$$\text{minimise } E(\mathbf{x}; \boldsymbol{\psi}) = \sum_{i=0}^{N-1} \|\mathbf{c}_{x_{i+1}} - \mathbf{c}_{x_i}\|_2 \tag{11}$$

$$\text{subject to } x_i \neq x_j \text{ for all } i \neq j \text{ and } x_i \in \{0, 1, ..., N-1\}, \tag{12}$$

where we have defined $x_N = x_0$ for convenience of notation. Our action space consists of so-called *2-opt* moves (Croes, 1958), which reverse contiguous segments of a tour. An example of a 2-opt move is shown in Figure 1. We have a two-stage architecture, like in Bin Packing, which selects the

Table 3: Comparison of Neural SA against different solvers and deep learning models on TSP. Extended version of this table is provided in the appendix. Lower is better.

|  | TSP20 | | | TSP50 | | | TSP100 | | | TSP200 | | | TSP500 | | |
|---|---|---|---|---|---|---|---|---|---|---|---|---|---|---|---|
|  | Cost | Gap | Time | Cost | Gap | Time | Cost | Gap | Time | Cost | Gap | Time | Cost | Gap | Time |
| CONCORDE | 3.836 | 0.00% | 48s | 5.696 | 0.00% | 2m | 7.764 | 0.00% | 7m | 10.70 | 0.00% | 38m | 16.54 | 0.00% | 7h58m |
| LKH-3 | 3.836 | 0.00% | 1m | 5.696 | 0.00% | 14m | 7.764 | 0.00% | 1h | 10.70 | 0.00% | 21m | 16.54 | 0.00% | 1h15m |
| SA | 3.881 | 1.17% | 10s | 5.943 | 4.34% | 1m | 8.341 | 7.43% | 6m | 11.98 | 11.96% | 30m | 20.22 | 22.25% | 2h35m |
| OURS (PPO) | 3.852 | 0.42% | 17s | 5.762 | 1.16% | 2m | 7.907 | 1.85% | 14m | 11.04 | 3.27% | 32m | 17.87 | 8.04% | 5h13m |
| OURS (ES) | 3.840 | 0.10% | 17s | 5.828 | 2.32% | 2m | 8.191 | 5.50% | 14m | 11.74 | 9.72% | 32m | 20.27 | 22.55% | 5h13m |
| GAT-T{1000} | 3.84 | 0.03% | 12m | 5.75 | 0.83% | 16m | 8.01 | 3.24% | 25m | - | - | - | - | - | - |
| Costa{500} | 3.84 | 0.01% | 5m | 5.72 | 0.36% | 7m | 7.91 | 1.84% | 10m | - | - | - | - | - | - |

start and end cities of the segment to reverse. Denoting $i$ as the start and $j$ as the end

$$\pi_{\theta,\phi}(\mathbf{a} = (i,j)|\mathbf{s}) = \pi_{\phi}(i|\mathbf{s})\pi_{\theta}(j|\mathbf{s},i) \tag{13}$$

$$\pi_{\theta}(i|\mathbf{s}) = \text{softmax}(\mathbf{z})_i, \qquad z_i = f_{\theta}([\mathbf{c}_{\mathbf{x}_{[i-1:i+1]}}, T]) \tag{14}$$

$$\pi_{\phi}(j|\mathbf{s},i) = \text{softmax}(\mathbf{z})_j, \qquad z_j = f_{\phi}([\mathbf{c}_{\mathbf{x}_{[i-1:i+1]}}, \mathbf{c}_{\mathbf{x}_{[j-1:j+1]}}, T]), \tag{15}$$

where $\mathbf{x}_{[i-1:i+1]}$ are the indices of city $i$ and its tour neighbours $i-1$ and $i+1$. WE use the standard policy: $f_{\theta}$ has architecture $7 \to 16 \to 1$ and $f_{\phi}$, $13 \to 16 \to 1$. We test on publically available TSP20/50/100 (Kool et al., 2018) with 10K problems each and generate TSP200/500 with 1K tours each. Results, in Table 11, show Neural SA improves on vanilla SA. Neural improvement heuristics methods, GAT-T{1000} (Wu et al., 2019b) and Costa{500} (de O. da Costa et al., 2020), designed for routing problems, outperform us for small TSPs, but are neck-and-neck for TSP100. Given Neural SA is not custom designed for TSPs, we view this as surprisingly good.

## 5 DISCUSSION

Neural SA is a general, plug-and-play method, requiring only definition of neighbourhoods and training problem instances. Our lightweight architecture shared for all problems we consider and, with a maximum of 384 parameters on TSP, 160 for Bin Packing, and 112 for Knapsack can achieve within a few percentage points or less of global minimia. It can be trained with any policy optimisation method making it highly extendable. We found no winner between PPO and ES, apart from on the TSP, where PPO excelled. We also observed PPO to converge $\sim 10\times$ faster than ES, but ES policies were more robust, such as when we switched to greedy sampling. Here the PPO policy was usually hurt, but ES was not. An interesting observation we made was that the acceptance rate over trajectories was problem dependent, always higher in Neural SA than vanilla SA. This contradicts conventional wisdom that it should be held at 0.44 throughout a rollout (Lam & Delosme, 1988).

While never achieving SOTA, this was never to be expected, given SA is an approximation meta-heuristic, but if left running long enough, results are guaranteed to improve, following from the convergence guarantees of SA. It is also worth pointing out we did little to no fine-tuning of the hyperparameters of SA, and yet achieved positive results even with an unsophisticated temperature cooling schedule. This demonstrates that besides improving convergence speeds, Neural SA is also more robust to variations in the optimisation dynamics of SA. In fact, our experiments show Neural SA generalises across different problem sizes and rollout lengths; a truly remarkable feat, as transfer learning is notoriously difficult in reinforcement learning and combinatorial optimisation. That makes Neural SA an attractive alternative in many applications where problem sizes and time constraints fluctuate, especially when computational power or energy consumption are limiting factors.

## 6 CONCLUSION

We presented *Neural SA*, neurally augmented simulated annealing, where the SA chain is a trajectory from an MDP. In this light, the proposal distribution could be interpreted as a policy, which could be optmised. This has numerous benefits: 1) accelerated convergence of the chain, 2) ability to condition the proposal distribution on side-information 3) we do not need groundtruth data to learn the proposal distribution, 4) our architectures are very lightweight and can be run on CPU unlike many contemporary ML4CO methods, 5) the method scales well due to its lightweight computational overhead, 6) we can train on small problems and generalise to much larger ones. Of these contributions, we believe 6 is the most impressive. Many ML4CO solutions currently use huge transformer-based models, with computation scaling quadratically per step with problem size; whereas, neural SA instead scales linearly. This enabled us to yield results on larger problem sizes.

Removing pairwise interactions in the model, however, could be its limiting factor. In all experiments we were not able to achieve the minimum energy, but could usually get within a percentage point. The model also has no in-built termination condition, neither can it provide a certificate on the quality of solutions found. There is still also the question of how to tune the temperature schedule, which we did not attempt in this work. These shortcomings are all points to be addressed in upcoming research. We are also interested in extending the framework to multiple trajectories, such as in parallel tempering Swendsen & Wang (1986) or genetic algorithms Holland (1992). For these, we would maintain a population of chains, which could exchange information.

# 7 REPRODUCIBILITY STATEMENT

Our method is outlined in Algorithm 1. In sections 4.1, 4.2, 4.3, and 4.4 we provide details of the exact models we use and the datasets. Each dataset was generated synthetically, apart from the TSP20, 50, and 100 datasets, which are publically available. We have reported all our numbers, with standard deviations, and provide precise training details in the appendix. As a precaution, we plan to release our code upon acceptance of this paper, for complete transparency.

# 8 ETHICS STATEMENT

Solving real world combinatorial optimisation problems can reduce waste, allocating scarce resources near optimally. This paper in particular introduces a lightweight method to improve upon simulated annealing, a popular combinatorial optimisation method, such that it can be targeted at a broad set of problems with ease. Where simulated annealing itself is deployed, for moral or likewise immoral purposes, is beyond the scope of this paper, but based on typical examples of combinatorial optimisation problems, our advances probably cause little downstream harm. The improvements in convergence speed we observed also reduce total FLOPs and therefore carbon footprint, a common problem with large-scale optimisations and deep learning models.

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

## A  ADDITIONAL EXPERIMENTAL INFORMATION AND RESULTS

Our code was implemented in Pytorch 1.9 (Paszke et al., 2017) and run in a standard machine equipped with a single GPU RTX2080. The randomly generated datasets used for testing can be recreated by setting the seed of Pytorch's random number generator to 0.

Table 4: Running Times

|        | Knapsack | | Bin Packing | |
|--------|----------|----------|----------|----------|
|        | Ours | OR-Tools | Ours | OR-Tools |
| 50N   | $0.74s$   | $< 0.01s$ | $1.14s$  | $54s$     |
| 100N  | $1.13s$   | $< 0.01s$ | $2.27s$  | $56s$     |
| 200N  | $3.01s$   | $< 0.01s$ | $4.79s$  | $\geq 1m$ |
| 500N  | $9.40s$   | $0.01s$   | $14.18s$ | $\geq 1m$ |
| 1000N | $38.68s$  | $0.02s$   | $37.67s$ | $\geq 1m$ |
| 2000N | $130.86s$ | $0.08s$   | $121.74s$| $\geq 1m$ |

### A.1  KNAPSACK PROBLEM

**Data**  We consider problems of different sizes, with KNAP$N$ consisting of $N$ items, each with a weight $w_i$ and value $v_i$ sampled from a uniform distribution, $w_i, v_i \sim \mathcal{U}_{(0:1)}$. Each problem has also an associated capacity, that is, the maximum weight the knapsack can comport. Here we follow (Bello et al., 2016) and set $C_{50} = 12.5, C_{100} = 25$ and $C_{200} = 25$. However, for larger problems we set $C_N = N/8$.

**Training**  We train *only on* KNAP50 with short rollouts of length $K = 100$ steps. The model is trained for 1000 epochs each of which is run on 256 random problems generated on the fly as described in the previous section. We set initial and final temperatures to $T_0$ and $T_K$, and compute the temperature decay as $\alpha = (T_K/T_0)^{\frac{1}{K}}$.

- **PPO**: Both actor and critic networks were optimised with Adam (Kingma & Ba, 2015) with learning rate of $1e-3$, weight decay of $1e-2$ and $\beta = (0.9, 0.999)$. $T_0 = 1$ and $T_K = 0.1$.
- **ES**: We use a population of 16 perturbations sampled from a Gaussian of standard deviation 0.05. Updates are fed into an SGD optimizer with learning rate 1e-3 and momentum 0.9. $T_0 = 1$ and $T_K = 0.1$.

**Testing**  We evaluate Neural SA on test sets of 1000 randomly generated Knapsack problems, while varying the length of the rollout. For each problem size $N$, we consider rollouts of length $K = N$, $K = 2N$, $K = 5N$ and $K = 10N$. The initial and final temperatures are kept fixed to $T_0 = 1$ and $T_K = 0.1$, respectively, and the temperature decay varies as function of $K$, $\alpha = (T_K/T_0)^{\frac{1}{K}}$.

We compare our methods against one of the dedicated solvers for knapsack in OR-Tools (Perron & Furnon) (KNAPSACK_MULTIDIMENSION_BRANCH_AND_BOUND_SOLVER). We also compare sampled and greedy variants of Neural SA. The former naturally samples actions from the proposal distribution while the latter always selects the most likely action.

Table 5: ES results on the Knapsack benchmark. Bigger is better. Comparison among rollouts of different lengths: 1, 2, 5 or 10 times the dimension of the problem.

| | Greedy | Sampled | | | | OR-Tools |
| | $\times 1$ | $\times 1$ | $\times 2$ | $\times 5$ | $\times 10$ | |
|---|---|---|---|---|---|---|
| KNAP50 | $16.59 \pm .00$ | $19.45 \pm .01$ | $19.70 \pm .00$ | $19.86 \pm .00$ | $19.95 \pm .00$ | **20.07** |
| KNAP100 | $31.15 \pm .00$ | $39.07 \pm .01$ | $39.49 \pm .01$ | $39.76 \pm .01$ | $39.90 \pm .01$ | **40.50** |
| KNAP200 | $55.96 \pm .00$ | $53.72 \pm .02$ | $55.21 \pm .02$ | $56.22 \pm .02$ | $56.58 \pm .01$ | **57.45** |
| KNAP500 | $135.92 \pm .00$ | $134.20 \pm .05$ | $137.89 \pm .03$ | $140.20 \pm .02$ | $141.01 \pm .03$ | **144.14** |
| KNAP1K | $259.20 \pm .00$ | $269.21 \pm .04$ | $276.48 \pm .05$ | $280.94 \pm .02$ | $282.46 \pm .03$ | **289.01** |
| KNAP2K | $489.02 \pm .00$ | $537.53 \pm .08$ | $551.92 \pm .07$ | $560.75 \pm .07$ | $563.75 \pm .02$ | **577.28** |

Table 6: PPO results on the Knapsack benchmark. Bigger is better. Comparison among rollouts of different lengths: 1, 2, 5 or 10 times the dimension of the problem.

| | Greedy | Sampled | | | | OR-Tools |
| | $\times 1$ | $\times 1$ | $\times 2$ | $\times 5$ | $\times 10$ | |
|---|---|---|---|---|---|---|
| KNAP50 | $19.46 \pm .00$ | $19.31 \pm .01$ | $19.43 \pm .01$ | $19.70 \pm .01$ | $19.87 \pm .01$ | **20.07** |
| KNAP100 | $38.79 \pm .00$ | $38.53 \pm .02$ | $38.92 \pm .02$ | $39.56 \pm .02$ | $39.92 \pm .01$ | **40.50** |
| KNAP200 | $47.90 \pm .00$ | $48.76 \pm .04$ | $51.13 \pm .03$ | $54.05 \pm .02$ | $55.65 \pm .01$ | **57.45** |
| KNAP500 | $117.51 \pm .00$ | $121.88 \pm .05$ | $128.78 \pm .03$ | $136.12 \pm .03$ | $139.66 \pm .02$ | **144.14** |
| KNAP1K | $234.34 \pm .00$ | $245.48 \pm .12$ | $259.80 \pm .05$ | $273.96 \pm .05$ | $280.39 \pm .01$ | **289.01** |
| KNAP2K | $464.92 \pm .00$ | $491.48 \pm .10$ | $520.61 \pm .06$ | $548.15 \pm .06$ | $560.40 \pm .02$ | **577.28** |

## A.2 BIN PACKING PROBLEM

**Data** We consider problems of different sizes, with BIN$N$ consisting of $N$ items, each with a weight (size) sampled from a uniform distribution, $w_i \sim \mathcal{U}_{(0:1)}$. Without loss of generality, we also assume $N$ bins, all with unitary capacity. Each dataset BIN$N$ in Tables 7 and 8 contains 1000 such random Bin Packing problems used to evaluate the methods at test time.

**Training** We train *only on* BIN50 with short rollouts of length $K = 100$ steps. The model is trained for 1000 epochs each of which is ran on 256 random problems generated on the fly as described in the previous section. We set initial and final temperatures to $T_0$ and $T_K$, and compute the temperature decay as $\alpha = (T_K/T_0)^{\frac{1}{K}}$.

- **PPO**: Both actor and critic networks were optimised with Adam (Kingma & Ba, 2015) with learning rate of 2e−4, weight decay of 1e−2 and $\beta = (0.9, 0.999)$. $T_0 = 1$ and $T_K = 0.1$.

- **ES**: We use a population of 16 perturbations sampled from a Gaussian of standard deviation 0.1. Updates are fed into an SGD optimizer with learning rate 1e-3 and momentum 0.9. $T_0 = 0.1$ and $T_K = 1e − 4$.

**Testing** We evaluate Neural SA on test sets of 1000 randomly generated Bin Packing problems, while varying the length of the rollout. For each problem size $N$, we consider rollouts of length $K = N$, $K = 2N$, $K = 5N$ and $K = 10N$. The initial and final temperatures are kept fixed to $T_0 = 1$ and $T_K = 0.1$, respectively, and the temperature decay varies as function of $K$, $\alpha = (T_K/T_0)^{\frac{1}{K}}$.

We compare Neural SA against First-Fit-Decreasing (FFD) (Johnson, 1973), a powerful heuristic for the Bin Packing problem, and against OR-Tools (Perron & Furnon) MIP solver powered by SCIP (Gamrath et al., 2020a). The OR-Tools solver can be quite slow on Bin Packing so we set a time out of one minute per problem so that we could complete

We also compare sampled and greedy variants of Neural SA. The former naturally samples actions from the proposal distribution while the latter always selects the most likely action.

Table 7: ES results on the Bin Packing benchmark. Lower is better.

| | Greedy ×1 | Sampled ×1 | ×2 | ×5 | ×10 | OR-Tools | FFD |
|---|---|---|---|---|---|---|---|
| Bin50 | 27.62±.00 | 27.43±.01 | 27.36±.01 | 27.29±.00 | 27.24±.01 | **26.71** | 27.10 |
| Bin100 | 53.80±.00 | 53.63±.00 | 53.54±.01 | 53.44±.01 | 53.38±.01 | 53.91 | **52.91** |
| Bin200 | 105.63±.00 | 105.78±.02 | 105.64±.01 | 105.51±.01 | 105.43±.01 | 109.19 | **104.25** |
| Bin500 | 259.09±.00 | 260.86±.03 | 260.65±.01 | 260.42±.02 | 260.27±.02 | 267.63 | **257.02** |
| Bin1K | 512.66±.00 | 517.87±.02 | 517.46±.02 | 517.08±.02 | 516.84±.01 | 1000* | **509.46** |
| Bin2K | **1017.88 ± .00** | 1030.66±.01 | 1029.89±.01 | 1029.11±.02 | 1028.67±.02 | 2000* | 1028.67 |

Table 8: PPO results on the Bin Packing benchmark. Lower is better.

| | Greedy ×1 | Sampled ×1 | ×2 | ×5 | ×10 | OR-Tools | FFD |
|---|---|---|---|---|---|---|---|
| Bin50 | 27.62 ± .00 | 27.83 ± .01 | 27.65 ± .01 | 27.41 ± .01 | 27.29 ± .01 | **26.71** | 27.10 |
| Bin100 | 53.80 ± .00 | 54.63 ± .02 | 54.15 ± .02 | 53.68 ± .01 | 53.47 ± .01 | 53.91 | **52.91** |
| Bin200 | 105.63 ± .00 | 108.04 ± .02 | 106.93 ± .02 | 106.08 ± .01 | 105.75 ± .01 | 109.19 | **104.25** |
| Bin500 | 259.08 ± .00 | 267.19 ± .05 | 264.14 ± .01 | 262.26 ± .02 | 261.61 ± .01 | 267.63 | **257.02** |
| Bin1K | 512.66 ± .00 | 531.17 ± .05 | 524.78 ± .03 | 521.32 ± .03 | 520.32 ± .02 | 1000* | **509.46** |
| Bin2K | **1017.88 ± .00** | 1058.74 ± .04 | 1045.36 ± .03 | 1038.77 ± .02 | 1037.15 ± .03 | 2000* | 1028.67 |

## A.3 TRAVELLING SALESPERSON PROBLEM (TSP)

**Data** We generate random instances for 2D Euclidean TSP by sampling coordinates uniformly in a unit square, as done in previous research (Kool et al., 2018; Chen & Tian, 2019; de O. da Costa et al., 2020). We assume complete graphs (fully-connected TSP), which means every pair of cities is connected by a valid route (an edge).

**Training** We train *only on* TSP20 with very short rollouts of length $K = 40$. Just like in the other problems we consider, we train using 256 random problems generated on the fly for each. We also maintain the same initial temperature and cooling schedule with $T_0 = 1$ and $\alpha = (T_K/T_0)^{\frac{1}{K}}$, but use lower final temperatures for the TSP. We set $T_K = 1e-2$ for PPO and $T_K = 1e-4$ for ES, which we empirically found to work best with the training dynamics of each of these methods.

- **PPO**: Both actor and critic networks were optimised with Adam (Kingma & Ba, 2015) with learning rate of $2e-4$, weight decay of $1e-2$ and $\beta = (0.9, 0.999)$. We trained PPO for 1000 epochs.

- **ES**: We use a population of 16 perturbations sampled from a Gaussian of standard deviation 0.1. Updates are fed into an SGD optimizer with learning rate 1e-3 and momentum 0.9, and trained for 10 000 epochs.

**Testing** We evaluate Neural SA on TSP20, TSP50 and TSP100 using the 10K problem instances made available in Kool et al. (2018). This allows us to directly compare our methods to previous research on the TSP. We also consider larger problem sizes, namely TSP200 and TSP500 to showcase the scalability of Neural SA. For each of these, we randomly generate 1000 instances by uniformly sampling coordinates in a 2D unit square. As for the other CO problems we study, we also compare sampled and greedy variants of Neural SA. The former naturally samples actions from the proposal distribution while the latter always selects the most likely action.

We compare Neural SA against standard solvers LKH-3 (Helsgaun, 2000) and Concorde (Applegate et al., 2006), which we have run ourselves. We also compare against the self-reported results of other Deep Learning models that have targeted TSP and relied on the test data provided by Kool et al. (2018): GCN (Joshi et al., 2019), GAT (Kool et al., 2018), GAT-T (Wu et al., 2019a), and the works of de O. da Costa et al. (2020) and Fu et al. (2021).

Note that Fu et al. (2021) also provide results for TSP200 and TSP500, but given that there is no public dataset for these problem sizes, it is hard to make a direct comparison to our results, especially regarding running times. They use a dataset of 128 instances, while we use 1000.

Table 9: PPO results on the TSP benchmark. Lower is better

|  | Greedy | Sampled | | | | LKH-3 | Concorde |
|  | ×1 | ×1 | ×2 | ×5 | ×10 |  |  |
|---|---|---|---|---|---|---|---|
| TSP20 | $4.937 \pm .001$ | $3.941 \pm .001$ | $3.898 \pm .001$ | $3.865 \pm .000$ | $3.852 \pm .000$ | **3.836** | **3.836** |
| TSP50 | $8.104 \pm .006$ | $5.916 \pm .002$ | $5.847 \pm .001$ | $5.789 \pm .000$ | $5.762 \pm .000$ | **5.696** | **5.696** |
| TSP100 | $11.764 \pm .012$ | $8.167 \pm .000$ | $8.052 \pm .001$ | $7.956 \pm .001$ | $7.907 \pm .001$ | **7.764** | **7.764** |

Table 10: ES results on the TSP benchmark. Lower is better

|  | Greedy | Sampled | | | | LKH-3 | Concorde |
|  | ×1 | ×1 | ×2 | ×5 | ×10 |  |  |
|---|---|---|---|---|---|---|---|
| TSP20 | $3.868 \pm .000$ | $3.868 \pm .001$ | $3.854 \pm .000$ | $3.844 \pm .000$ | $3.840 \pm .000$ | **3.836** | **3.836** |
| TSP50 | $6.020 \pm .002$ | $6.022 \pm .002$ | $5.947 \pm .001$ | $5.871 \pm .000$ | $5.828 \pm .001$ | **5.696** | **5.696** |
| TSP100 | $8.659 \pm .003$ | $8.660 \pm .002$ | $8.477 \pm .001$ | $8.298 \pm .002$ | $8.191 \pm .002$ | **7.764** | **7.764** |

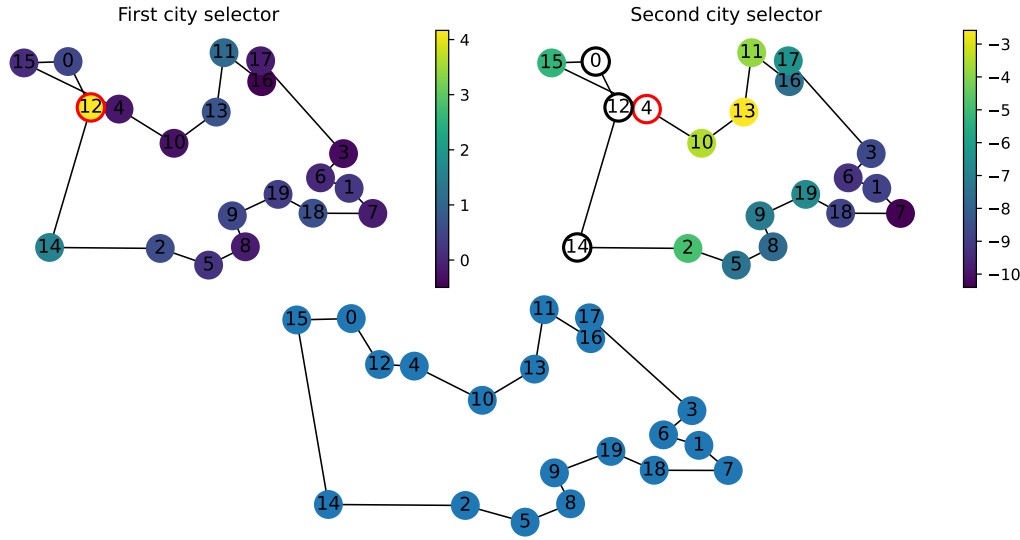

Figure 6: Policy for the Travelling Salesperson Problem. At each step, an action consists of selecting a pair of cities $(i, j)$, one after the other. The figure depicts a TSP problem layed out in the 2D plane, with the learnt proposal distribution over the first city $i$ in the left, and in the right, the distribution over the second city $j$, given $i = 12$. We mask out and exclude the neighbours of $i$ (0 and 14) as candidates for $j$ because selecting those would lead to no changes in the tour. It is clear the model has a strong preference towards a few cities, but otherwise the probability mass is spread almost uniformly among the other nodes. However, once $i$ is fixed, Neural SA strongly favours nodes $j$ that are close to $i$. That is a desirable behaviour and even features in popular algorithms like LKH-3 (Helsgaun, 2000). That is because a 2-opt move $(i, j)$ actually adds edge $(i, j)$ to the tour, so leaning towards pairs of cities that are close to each other is more likely to lead to shorter tours.

Table 11: Comparison of different TSP solvers on the 10K instances for TSP20/50/100 provided in Kool et al. (2018).

| | TSP20 | | | TSP50 | | | TSP100 | | | TSP200 | | | TSP500 | | |
| --- | --- | --- | --- | --- | --- | --- | --- | --- | --- | --- | --- | --- | --- | --- | --- |
| | Cost | Gap | Time | Cost | Gap | Time | Cost | Gap | Time | Cost | Gap | Time | Cost | Gap | Time |
| CONCORDE | 3.836 | 0.00% | 48s | 5.696 | 0.00% | 2m | 7.764 | 0.00% | 7m | 10.70 | 0.00% | 38m | 16.54 | 0.00% | 7h58m |
| LKH-3 | 3.836 | 0.00% | 1m | 5.696 | 0.00% | 14m | 7.764 | 0.00% | 1h | 10.70 | 0.00% | 21m | 16.54 | 0.00% | 1h15m |
| SA | 3.881 | 1.17% | 10s | 5.943 | 4.34% | 1m | 8.341 | 7.43% | 6m | 11.98 | 11.96% | 30m | 20.22 | 22.25% | 2h35m |
| NEURAL SA PPO | 3.852 | 0.42% | 17s | 5.762 | 1.16% | 2m | 7.907 | 1.85% | 14m | 11.04 | 3.27% | 32m | 17.87 | 8.04% | 5h13m |
| NEURAL SA ES | 3.840 | 0.10% | 17s | 5.828 | 2.32% | 2m | 8.191 | 5.50% | 14m | 11.74 | 9.72% | 32m | 20.27 | 22.55% | 5h13m |
| GCN | 3.86 | 0.60% | 6s | 5.87 | 3.10% | 55s | 8.41 | 8.38% | 6m | - | - | - | - | - | - |
| GCN + Beam Search | 3.84 | 0.01% | 12m | 5.70 | 0.01% | 18m | 7.87 | 1.39% | 40 m | - | - | - | - | - | - |
| GAT | 3.84 | 0.08% | 5 m | 5.73 | 0.52% | 24m | 7.94 | 2.26% | 1 h | - | - | - | - | - | - |
| GAT-T {1000} | 3.84 | 0.03% | 12m | 5.75 | 0.83% | 16m | 8.01 | 3.24% | 25m | - | - | - | - | - | - |
| GAT-T {3000} | 3.84 | 0.00% | 39m | 5.72 | 0.34% | 45 m | 7.91 | 1.85% | 1 h | - | - | - | - | - | - |
| GAT-T {5000} | 3.84 | 0.00% | 1 h | 5.71 | 0.20% | 1 h | 7.87 | 1.42% | 2 h | - | - | - | - | - | - |
| Da Costa et al. {500} | 3.84 | 0.01% | 5m | 5.72 | 0.36% | 7m | 7.91 | 1.84% | 10m | - | - | - | - | - | - |
| Da Costa et al. {1000} | 3.84 | 0.00% | 10m | 5.71 | 0.21% | 13m | 7.86 | 1.26% | 21 m | - | - | - | - | - | - |
| Da Costa et al. {2000} | 3.84 | 0.00% | 15m | 5.70 | 0.12% | 29m | 7.83 | 0.87% | 41m | - | - | - | - | - | - |
| Fu et al. | 3.84 | 0.00% | 2m | 5.69 | 0.01% | 9m | 7.76 | 0.03% | 15m | 10.81 | 0.88% | 3m | 16.96 | 2.96% | 6m |

