# OpenReview forum: "Neural Simulated Annealing"
_ICLR.cc/2022/Conference — ICLR 2022 Submitted_

### Official Review · Reviewer_HRDu · 2021-10-24

**Correctness:** 4
**Technical Novelty And Significance:** 3
**Empirical Novelty And Significance:** 3
**Recommendation:** 8
**Confidence:** 4

**Main Review:**

**Strengths**

I believe the paper has the following strengths:
- A modular and practical improvement on a traditional optimization technique (SA), which demonstrates clear improvement on the optimization benchmarks provided.
- A detailed and thorough description of the method, including how SA can interact with reinforcement learning in a scalable manner.
- A thorough analysis of the results provided by the Neural Simulated Annealing policy for the Knapsack and Bin Packing problem, which showcase that the policy is behaving as expected. This strengthens the claim that Neural Simulated Annealing is modular, but significant, improvement on SA.

**Weaknesses**

I think that the paper could be improved by:
- Providing a discussion and analysis of the compute cost incurred by Neural Simulated Annealing compared to traditional SA. While it appears that Neural Simulated Annealing provides performance improvements, it would be good to understand the costs of those performance improvements.
- A discussion of whether the proposal distribution can be determined other than using softmax across the entire SA chain. Would good advantages and disadvantages of other methods?
- A discussion on what the authors think would be needed to achieve better results on the benchmarks where SOTA wasn't achieved, such as the TSP. Is it mainly a question of letting the algorithm run longer or are there other things that can be done to arrive at better solutions?
- Stating the size of the combinatorial search space for each experiment that is performed. E.g. Knap200 has a search space of X, which scales to Y in Knap2K because of Z. This will add further clarity to how the method scales to more challenging settings.

While I don't think this is necessary at this moment, I think it would have been nice to have the following as well:
- Results on practical, "real-world" problem where the advantages of Neural Simulated Annealing could be shown. TSP is a good approximation to many COs found in real-world settings, but it would have been nice to see the algorithm applied to other examples as well.
- An ablation of the state definition to see what information may or may not be critical for NSA to tackle the problem effectively.
- A discussion on how advances in RL, such as multi-task learning or meta-learning, could be used to potentially generalize NSA across different tasks.


**Summary Of The Paper:**

The paper introduces a new method called Neural Simulated Annealing, which builds upon traditional Simulated Annealing (SA) by introducing a lightweight neural network as a policy that determines the proposal distribution for SA. The neural network policy is trained by a reinforcement learning (RL) mechanism, which relies on re-formulating the Markov chain of the underlying combinatorial optimization problem as a Markov Decision Process (MDP).  The authors outline how their re-formulation fits into the definitions of SA and MDPs in Section 2 and Section 3, defining the state, action, transition function and reward function (immediate or primal) be aligned to enable RL to train the neural network policy while also being amendable to the general SA optimization structure. The authors then provide an argument for why their method has the same convergence guarantees as traditional SA.

Following the general definition of their method and their algorithm, including the definition of the neural network policy, the authors then show results for 4 tasks: Rosenbrock's function, the Knapsack problem, Bin Packing and the Traveling Salesman problem applying their method, traditional SA and relevant methods from the literature. The results from the authors generally show improvement of their method compared to traditional SA and outperformance compared to other methods in some, but not all, of the cases. The authors also provide an analysis of the results of Neural Simulated Annealing for the Knapsack and Bin Packing problem claiming that the behavior of the policy generally matches an intuitive behavior expected for solving those problems. Finally, the authors provide a discussion of their overall results and some ideas for future work.

**Summary Of The Review:**

I think that the paper presents a detailed and thorough framework for improving simulated annealing with a good set of experimental evidence to support the claims. I hope that the feedback provided will help the authors further improve the current draft during the discussion phase, and am open to adjusting my score based on the discussion.

I am also happy about the reproducibility statement provided by the authors, which outlines that the code will be published upon formal publication of the paper.

----- Update During Discussion Period -----

The authors addressed many of my questions and reservations in a satisfying manner. Assuming the authors make the changes they outline in their response, I think the paper will be significantly improved and merit acceptance. I am increasing my score to reflect this and will follow up on the technical discussion.

---

> ### Author Response · Authors · 2021-11-16
> **Response to Reviewer HRDu**
>
> We thank the reviewer for the interesting comments and suggestions. We expand the discussion on some of them below.
>
> > Providing a discussion and analysis of the compute cost incurred by Neural Simulated Annealing compared to traditional SA. While it appears that Neural Simulated Annealing provides performance improvements, it would be good to understand the costs of those performance improvements.
>
> Even though Neural SA does not require expensive neural architectures, evaluating the proposal distribution at each step incurs an additional computational cost on top of vanilla SA. However, the improvement in convergence speed achieved by Neural SA more than compensates for that extra computational expense. Thanks for bringing that up. Concretely we will add a complexity analysis in the paper and a comparison plot of convergence speed, like Figure 5.a), but with wall clock time on the horizontal axis.
>
> > A discussion of whether the proposal distribution can be determined other than using softmax across the entire SA chain. Would good advantages and disadvantages of other methods?
>
> We were not entirely clear what was meant by this comment, would you mind elaborating? Nonetheless, Neural SA defines a distribution over the action space. For Knapsack, Bin Packing and TSP, we defined a discrete action space for which a categorical distribution (which relies on a softmax operation) is a natural choice. Conversely, for the Rosenbrock function, the action space is actually continuous and we parametrise the proposal as a normal distribution.
>
> > A discussion on what the authors think would be needed to achieve better results on the benchmarks where SOTA wasn't achieved, such as the TSP. Is it mainly a question of letting the algorithm run longer or are there other things that can be done to arrive at better solutions?
>
> Running Neural SA for longer is guaranteed to improve results, but there are other aspects of the algorithm that could be fine-tuned for a particular problem. We list some of those below but emphasise that we aimed for the simplest of model designs with little to no fine-tuning, which goes to show the ease with which Neural SA can be applied to new problems with little work.
> - *Temperature schedule*. SA is highly dependent on the temperature schedule and, while Neural SA mitigates that by relying on a smarter proposal distribution, the temperature schedule could still be optimised or even learned, which is an interesting step for future research.
> - *Feature and action space engineering*. We aimed for simplicity and chose bare-bones and intuitive features and action spaces. This is in sharp contrast to other deep learning approaches to TSP, for instance, where features are carefully designed or further processed by expensive architectures to build expressive representations.
> - *More expressive architectures*. Our architectures are fairly simple, and while they do significantly improve over vanilla SA, it is reasonable to assume that more powerful architectures could yield better results, even if at the cost of slower training and inference times. For instance, one could include attention mechanisms to model the interaction between different dimensions of a problem, something that is currently ignored by our simplistic equivariant architectures.
>
> > Stating the size of the combinatorial search space for each experiment that is performed. E.g. Knap200 has a search space of X, which scales to Y in Knap2K because of Z. This will add further clarity to how the method scales to more challenging settings.
>
> Thanks for the suggestion, we will include a discussion on the size of the search spaces in the paper.
>
> > - Results on practical, "real-world" problem where the advantages of Neural Simulated Annealing could be shown. TSP is a good approximation to many COs found in real-world settings, but it would have been nice to see the algorithm applied to other examples as well.
> > - An ablation of the state definition to see what information may or may not be critical for NSA to tackle the problem effectively.
> > - A discussion on how advances in RL, such as multi-task learning or meta-learning, could be used to potentially generalize NSA across different tasks.
>
> Those are all useful suggestions. Thanks for bringing them up. We will consider expanding the paper in those directions in the final version.

---

> > ### Comment · Reviewer_HRDu · 2021-11-20
> > **Further Discussion**
> >
> > Thank you for your thorough reply to my comments and for committing to various additions that should improve the strength of the paper.
> >
> > I wanted to follow up on the question regarding softmax. Based on my understanding, the softmax function is pretty useful for your application because it transforms the logit based output of the SA chain to something that looks more like a probability distribution, which you can then use as your proposal distribution. My question relates to whether you think of other ways to arrive at a proposal distribution based on the logit outputs and what potential advantages or disadvantages of those methods would be compared to the softmax based approach you current have.

---

> > > ### Author Response · Authors · 2021-11-22
> > > **Response to Reviewer HRDu (2)**
> > >
> > > We would like to thank the reviewer for taking the time to evaluate our rebuttal and for having improved the score of the paper.
> > >
> > > > I wanted to follow up on the question regarding softmax. Based on my understanding, the softmax function is pretty useful for your application because it transforms the logit based output of the SA chain to something that looks more like a probability distribution, which you can then use as your proposal distribution. My question relates to whether you think of other ways to arrive at a proposal distribution based on the logit outputs and what potential advantages or disadvantages of those methods would be compared to the softmax based approach you current have.
> > >
> > > Thanks for the clarification. This is an interesting point, especially considering that a naive softmax implementation might be problematic for large numbers of categories. At the end of the day, the logits already give us a valid (but unnormalised) proposal distribution for discrete action spaces that we can easily sample from, for instance using the Gumbel-Max trick—the softmax is only really needed to train with PPO, where we do need normalised probability estimates.
> > >
> > > We have not yet considered distributions other than a straightforward categorical distribution, but one possible avenue to improve on this could be to take the structure of the action space into account. For instance, in the TSP it might make sense to define a hierarchical structure based on the current tour and spatial locations of cities, and sample in stages: first sample a region of the space to operate on and then gradually ‘zoom in’ until sampling a single city.
> > >
> > > Another generalisation of our technique would be to deploy it to problems with structured outputs where solutions themselves are structured, such as trees or graphs. In this case we would need to be able to sample from a distribution over these structured objects. We would not need to know the probabilities of the samples if training with Evolution Strategies. Such sampling strategies exist, such as in "Gradient Estimation with Stochastic Softmax Tricks" (Paulus et al., 2020), where the authors note that you can sample such structured objects by solving a constrained convex program.

---

### Official Review · Reviewer_4KAJ · 2021-10-30

**Correctness:** 2
**Technical Novelty And Significance:** 3
**Empirical Novelty And Significance:** 2
**Recommendation:** 5
**Confidence:** 5

**Main Review:**

Pros:
=

1. The idea is interesting and sound. Simulated annealing can be viewed as Markov decision process naturally, which motivates the use of reinforcement learning.

2. Although there has been some studies for hybridizing SA and learning methods, the authors have distinguished their work from existing studies clearly.

3. The experiments on four optimization problems demonstrate that the proposed neural SA works better than the SA alone.


Cons
=

1. Although the proposed neural SA improves over SA, its performance is still not comparable to other baselines (even a greedy method). I think this is the main weakness of this paper. Further, the authors claim that  "We demonstrate superior performance to off-the-shelf CO tools on the Knapsack, Bin Packing, and Travelling Salesperson problems, in terms of solution quality and wall-clock time." However, this is clearly NOT the case by looking at the experimental results.

2. There is no runtime comparison in Table 1 (Knapsack problem) and Table 2 (for Bin Packing problem). Hence, there is no way to assess how fast the proposed method compared to other baselines.

3. OR-Tools should be tested for the traveling salesman problem as well, since it is a general method and has been tested on the other two problems.

**Summary Of The Paper:**

The paper proposes a new Neural Simulated Annealing approach to solve optimization problems. Specifically, reinforcement learning is used to optimize the proposal distribution in Simulated Annealing (SA). The proposed neural SA is empirically shown to be more effective than the classic SA without learning on four optimization problems. Although the neural SA outperforms SA, its performance is still not comparable to  other baselines.


**Summary Of The Review:**

I like the idea of viewing simulated annealing as the Markov decision process, and using reinforcement learning to enhance simulated annealing. My major concern is that the proposed neural simulated annealing is clearly not competitive with other baselines.

---

> ### Author Response · Authors · 2021-11-16
> **Response to Reviewer 4KAJ**
>
> We thank the reviewer for the useful feedback and address the most important concerns below.
>
> > Although the proposed neural SA improves over SA, its performance is still not comparable to other baselines (even a greedy method). I think this is the main weakness of this paper.
>
> The key contribution of Neural SA is that it is a simple, straightforward method, which performs well across a number of fairly different CO problems. Other than the choice of features we use at the input of each proposal network, there is very little tailoring that we actually do to each problem. So indeed we do not achieve SOTA on each problem. The utility of our method is its simplicity. Neural SA a fairly general approach that is easy to define and train on new combinatorial optimisation problems and yet delivers solid results. In sharp contrast to our method, the baselines we compare against are highly fine-tuned to each of the problems we considered. This becomes important in practical applications that either do not match traditional CO problems or simply fail to meet the assumptions of these dedicated baselines. In those scenarios, Neural SA is a promising alternative that is easy to use and still produces good results.
>
> We also believe the insight that the optimised Neural SA can get very close to many SOTA neural methods, but with just ~100 parameters, brings a new perspective to the discussion on where the community should be placing effort in ML4CO. This is namely that neural augmentation and other methods, which preserve the structure of other classical solvers can be very effective, with minimal learnable components.
>
> > Further, the authors claim that "We demonstrate superior performance to off-the-shelf CO tools on the Knapsack, Bin Packing, and Travelling Salesperson problems, in terms of solution quality and wall-clock time." However, this is clearly NOT the case by looking at the experimental results.
>
> We agree the phrasing of this sentence is slightly ambiguous, and we shall certainly change it. We will reword the abstract to indicate exactly which off-the-shelf CO tools we were referring to (ORTools) and make the limitations of our methods more clear. Thanks for spotting this.
>
> > There is no runtime comparison in Table 1 (Knapsack problem) and Table 2 (for Bin Packing problem). Hence, there is no way to assess how fast the proposed method compared to other baselines.
>
> Due to space constraints we left runtime comparisons to the appendix, where they are reported in Table 4. We will include runtime information in the main text for clarity.
>
> > OR-Tools should be tested for the traveling salesman problem as well, since it is a general method and has been tested on the other two problems.
>
> That is a good suggestion. We will include a comparison against OR-Tools.

---

### Official Review · Reviewer_C2r3 · 2021-11-02

**Correctness:** 3
**Technical Novelty And Significance:** 3
**Empirical Novelty And Significance:** 2
**Recommendation:** 5
**Confidence:** 4

**Main Review:**

**Strengths**

- The paper is broadly well written, but I have provided a list of minor typo’s in Errata the author’s may wish to tidy.

- I believe the algorithm is novel and well motivated.  There have been several methods that combine SA and RL, however the authors provide a good review in Sec 2.1 and I agree that directly learning the proposal distribution is, to the best of my knowledge, novel in the context discussed within the paper. (With regards to methods that instead learn a temperature schedule, I feel [1] and [2] are a couple of recent works that may also merit inclusion in the review.)  However, I do feel the authors should avoid claiming that one of their contributions is presenting “simulated annealing as a Markov decision process, bringing it into the realm of reinforcement learning”, as previous works, including [1-2], already frame SA as an MDP and apply RL.

- That Neural SA improves over vanilla SA so significantly, despite the very modest network sizes, is impressive.

**Weakenesses**

- The author’s emphasise the ability of neural SA to generalise to larger problem sizes that those on which it was trained, remarking that “this type of transfer learning is rare in ML4CO” (Sec 4) and that the demonstrated generalisation is “a truly remarkable feat, as transfer learning is notoriously difficult [in RL and CO]” (Sec 5). Generalisation to unseen instances is indeed a highly sought after feature of CO heuristics, however I do not feel these claims are suitably justified.  Many ML heuristics can be applied to larger problems on which they were solved, including many of those referenced in the paper (for example Dai et al., 2017, Kool et al., 2018, Bresson & Laurent, 2021, Gasse et al 2019, Gupta et al 2020, Kool et al., 2021, Fu et al 2021 to select a few from the introduction).  Moreover, as SA can also generalise to larger instances (in the sense that, if we hand-craft a proposal policy and temperature schedule on small instances, it would not immediately fail on larger ones) I would suggest the author’s either (i) justify why Neural SA might naively be expected to not generalise (as to me, I would of been more surprised if it hadn’t) or (ii) demonstrate that the generalisation performance is significantly better than other heuristics or RL baselines.  If neither of these is possible I believe it would be appropriate to soften the discussion of generalisation, as it is not a unique feature of Neural SA, or demonstrated to be particularly exceptional.

- The information presented in the experimental results could be improved.  For the Bin-Packing problem (Table 2), Neural SA is shown to outperform SCIP however SCIP was given a 1 minute time limit and there is not reference to the run time of Neural SA.  If Neural SA was better within a commensurate time-budget, this would strengthen the reported result significantly.  If this is not the case, then the wall-clock time remains important context.  The TSP results in Table 3 are a subset of the full results in Table 11 (found in the appendix), however the strongest baseline (Fu et al) is not reported in the main text.  As such, it feels like the author’s claim to be “neck-and-neck” for TSP100 given Table 3, falsely gives the impression of Neural SA approaching SOTA RL performance.  I would ask the authors to either rectify the omission and update the text accordingly, or justify why Fu et al is not a better choice of baseline than those presented in Table 3.

- I believe the core idea of combining learnt local-improvement operators with a Metropolis-Hasting’s search procedure is very interesting, however experimental results are underwhelming.  Vanilla SA is poor (compared to other baselines) on the CO problems presented, and whilst learning the proposal step improves performance, by the author’s admission it is not state-of-the-art.  It would be interesting to select a problem where SA is already very strong, for example Ising model’s/Max-Cut problems, and see if Neural SA can still improve performance.  Alternatively, investigating wether the addition of a Metropolis-Hastings step to existing strong ML4CO algorithms that learn local improvement operators (e.g. [3]), would help to determine whether Neural SA’s advantage over SA arises from simply learning a better local-improvement step in general, or it the policy is bespoke to SA.

[1] “Reinforcement Learning Enhanced Quantum-inspired Algorithm for Combinatorial Optimization”, arXiv:2002.04676 (2020)

[2] "Finding the ground state of spin Hamiltonians with reinforcement learning”, Nat Mach Intell 2, 509–517 (2020).

[3] "Exploratory Combinatorial Optimization with Reinforcement Learning”, arXiv:1909.04063 (2019).

**Errata**

A few minor points that do not require and response from the authors.

- First paragraph of pg. 7: “groundtruth” - I think “ground truth” is more standard.
- First paragraph of pg. 7: “Results in Table 1, show that…” - I do not think there should be a comma after ‘Table 1’.
- First paragraph of pg. 8: “We sample from the policy ancestrally,” - should this be autoregressively?
- First paragraph of pg. 9: “WE use the” - erroneous capitalisation.

**Summary Of The Paper:**

The paper introduces Neural Simulated Annealing (Neural SA), a heuristic for general combinatorial optimisation (CO) problems.  Considering SA as a two stage process where, (i) given an initial state and new state is proposed and (ii) the new state is probabilistically accepted/rejected depending on the solution quality and current temperature (via a standard Metropolis-Hastings step), Neural SA parameterises the proposal step, (i), with a neural network that outputs a distribution over possible perturbations.  Framing the overall optimisation trajectory as a Markov Decision Process, the proposal policy can be trained using standard RL algorithms (both PPO and evolutionary strategies are used).  Experiments on four tasks — a simple proof-of-concept minimisation and three canonical CO problems — show that the learnt proposal policy significantly outperforms random perturbations (‘Vanilla SA’).  Whilst Neural SA does not surpass SOTA ML algorithms on the CO problems, it still provides strong performance with very small network sizes and the learnt policy can generalise to larger instances than those on which it was trained.

**Summary Of The Review:**

Whilst I believe the core idea of Neural SA is interesting and novel, I do not feel that it is sufficiently explored or demonstrates strong enough empirical performance to recommend acceptance at this time.  This is primarily because the algorithm neither achieves SOTA performance on considered problems, or demonstrates unique benefits not found in other systems (I believe the ability to generalise to larger problems was intended to address the second point, however, for the reasons discussed above, I am not convinced by this argument).  In my comments above, I have tried to present a few suggestions for how the paper could be extended to address these concerns - and would of course be open to pushback from the authors - however in it’s current form, my opinion is that this is just below the acceptance threshold.

---

> ### Author Response · Authors · 2021-11-16
> **Response to Reviewer C2r3**
>
> We thank the reviewer for the useful feedback and comments, which we address below.
>
> > [...] the authors should avoid claiming that one of their contributions is presenting “simulated annealing as a Markov decision process, bringing it into the realm of reinforcement learning”, as previous works, including [1-2], already frame SA as an MDP and apply RL.
>
> Thanks for the references, we will include them in the discussion about related work. It is important to point out that both Beloborodov et al. 2020 [1] and Mills et al. 2020 [9] do not frame SA itself as an MDP but rather use RL to optimise the hyperparameters of SA: the regularisation function in [1] and the temperature schedule in [9]. This is a key distinction that we will discuss further in the paper.
>
> ### Regarding generalisation
>
> Most of the models listed are based on graph neural networks or attention mechanisms that indeed accept inputs of different sizes. Indeed Fu et al. 2021, Gupta et al. 2020 and Gasse et al. 2019 do evaluate their models on different problem sizes, but that does not mean generalising to larger problems is trivial. For instance, to achieve generalisation to larger instances Fu et al. 2021 use a complex pipeline that feeds scores produced by a graph neural network into a Monte Carlo Tree Search algorithm. In contrast, Neural SA achieves similar results with a simple architecture and pipeline, which is noteworthy.
>
> Moreover, it has been observed that while many of the models in the literature can indeed accommodate different problem sizes, they do not perform well on problems larger than the ones seen during training. This is regarded as a main challenge in the field [4, 5, 6]. A very good manuscript on this problem is "Learning TSP Requires Rethinking Generalization" [6]. In their abstract they state "While state-of-the-art Machine Learning approaches perform closely to classical solvers when trained on trivially small sizes, they are unable to generalize the learnt policy to larger instances of practical scales". It is to this phenomenon that we were referring, in the sections that you have quoted and we shall clarify this in the text so that readers are aware of this issue.
>
> > Moreover, as SA can also generalise to larger instances [...] I would suggest the author’s either (i) justify why Neural SA might naively be expected to not generalise or (ii) demonstrate that the generalisation performance is significantly better than other heuristics or RL baselines.
>
> With enough time SA will always converge to the global minimiser. The same goes for Neural SA, which we showed inherits the convergence properties of standard SA. The main question we address is how fast. In general the larger the problem, the slower that convergence speed. When we talk about generalisation, what we are referring to is how well the acceleration in convergence speed transfers across problem size. Now when we compare to other methods in the literature things get complicated, because of the difference in the use of construction heuristics [6, 7, 8] and improvement heuristics [2, 3, 4]. Construction heuristics do not share the same notion of generalisation as we do, because they are never guaranteed to converge to an optimal solution, no matter how long you leave them. But we can level things up, if we consider the quality of the best solution after N steps of an improvement heuristic. It is in this particular sense that we do not expect vanilla SA to generalise across problem size, because we do not expect the convergence speed to remain the same across problem size.
>
> That said, we agree with the reviewer that we should do a better job in explaining this nuance. It is indeed a very interesting point. We are thankful for that feedback and will revise the text accordingly.
>
> ### Regarding runtimes for the Bin-Packing problem
>
> We report runtimes in the appendix, Table 4. Neural SA takes significantly less than one minute for all bin packing problems with up to 1000 bins. It does take a bit longer for 2000 bins, and thus we will update the timeout in OR-Tools for Bin2000 accordingly. Thanks for spotting that.
>
> ### Regarding TSP results (Table 3)
>
> The results for other deep learning methods are presented as reported by their authors, as all papers evaluate their methods on the 10000 instances made available in [6]. We selected the methods in Table 3 to best match the running times of Neural SA and omitted results by Fu et al. because it was not entirely clear whether they used the exact same instances as in [7]. However, we do agree this might lead to confusion and will include the results of Fu et al. in Table 3 and update the discussion to make the points mentioned above more clear.

---

> > ### Author Response · Authors · 2021-11-16
> > **(continued)**
> >
> > > I believe the core idea of combining learnt local-improvement operators with a Metropolis-Hasting’s search procedure is very interesting, however experimental results are underwhelming. Vanilla SA is poor (compared to other baselines) on the CO problems presented, and whilst learning the proposal step improves performance, by the author’s admission it is not state-of-the-art.
> >
> > Simulated annealing is a rather popular meta-heuristic with numerous variations, and performance improvements are usually obtained by fine-tuning a number of design choices and hyperparameters for a specific problem [10]. This could have been done for both vanilla SA and Neural SA (most of the variants would also be compatible with our method), but as our goal was to study the benefits of neural augmentation SA and not necessarily to achieve state-of-the-art in the individual problems we consider, we did not perform such extensive fine-tuning either for vanilla or Neural SA.
> >
> > > It would be interesting to select a problem where SA is already very strong, for example Ising model’s/Max-Cut problems, and see if Neural SA can still improve performance. Alternatively, investigating wether the addition of a Metropolis-Hastings step to existing strong ML4CO algorithms that learn local improvement operators (e.g. [3]), would help to determine whether Neural SA’s advantage over SA arises from simply learning a better local-improvement step in general, or it the policy is bespoke to SA.
> >
> > Those are interesting suggestions and promising avenues to extend our work. Indeed, understanding the exact mechanism by which Neural SA improves on standard SA, as you suggest, might benefit future research on ML4CO. Here we conjecture that the learnt policy is doing more than simply proposing local-improvements and actually aims at speeding up the convergence of SA as a whole. That is because we did observe that optimising the policy over the entire horizon of the optimisation process is important for the overall performance.
> >
> > We also agree that a Metropolis-Hasting step might prove beneficial to other ML4CO approaches, as it might allow a finer control over exploration via the temperature parameter. However, we believe this is beyond the scope of this work, which focuses on neurally augmenting SA.
> >
> > [1] Beloborodov, D., et al. "Reinforcement learning enhanced quantum-inspired algorithm for combinatorial optimization." Machine Learning: Science and Technology 2.2 (2020): 025009.
> >
> > [2] Chen, X., et al. "Learning to perform local rewriting for combinatorial optimization." NeurIPS 2019.
> >
> > [3] da Costa, P.R.O., et al. "Learning 2-opt heuristics for the traveling salesman problem via deep reinforcement learning." ACML2020, PMLR 129:465-480.
> >
> > [4] Fu, Z., et al. "Generalize a Small Pre-trained Model to Arbitrarily Large TSP Instances." AAAI 2021.
> >
> > [5] Gupta, P., et al. "Hybrid Models for Learning to Branch." NeurIPS 2020.
> >
> > [6] Joshi, C.K., et al. "Learning TSP requires rethinking generalization." arXiv preprint arXiv:2006.07054 (2020).
> >
> > [7] Kool, W., et al. "Attention, Learn to Solve Routing Problems!." ICLR 2018.
> >
> > [8] Kool, W., et al. "Deep Policy Dynamic Programming for Vehicle Routing Problems." arXiv preprint arXiv:2102.11756 (2021).
> >
> > [9] ​​Mills, K., et al. "Finding the ground state of spin Hamiltonians with reinforcement learning." Nature Machine Intelligence 2.9 (2020): 509-517.
> >
> > [10] Franzin, A., and Stützle, T. "Revisiting simulated annealing: A component-based analysis." Computers & operations research 104 (2019): 191-206.

---

> > ### Comment · Reviewer_C2r3 · 2021-11-21
> > **rebuttal**
> >
> > Thank you for the response and clarifications.  I wanted to pick up on a couple of points and clarify my concerns, as I do not feel these have not been adequately addressed.
> >
> > >It is important to point out that both Beloborodov et al. 2020 [1] and Mills et al. 2020 [9] do not frame SA itself as an MDP but rather use RL to optimise the hyperparameters of SA: the regularisation function in [1] and the temperature schedule in [9]. This is a key distinction that we will discuss further in the paper.
> >
> > I agree with the authors that these works do not share the MDP formulation presented in this work (I did not, and do not, question the novelty of Neural SA itself).  However my concern with the original statement, was that SA has already been combined with RL — therefore the claim that their work is “bringing it [simulating annealing] into the realm of reinforcement learning” is not correct.
> >
> > Regarding generalisation
> >
> > Again, I agree that generalisation is not trivial (in ML4CO or, indeed, ML more broadly).   I also see the distinction the authors are drawing between algorithms with convergence guarantees (where we can only consider generalisation as “how quickly will we find the solution”) and heurisitcs (where we might ask “how quickly will it find a solution of a given quality”/“how good of a solution will we find in a certain time limit”) — however, this feels beside the point as all framings are practical metrics of an algorithms utility, trading off speed and solution quality.  My comment was that the authors state that the generalisation that they present is “a truly remarkable feat, as transfer learning is notoriously difficult [in RL and CO]”, but fail to justify this very strong claim.  At present, I do not feel the authors response or the promise of discussing the nuances of generalisation address this concern.  Other algorithms can generalise across problem sizes/instances with vary levels of performance drop, as does Neural SA, so it is natural to ask why their performance is “a truly remarkable feat”.
> >
> > Runtimes for Bin-Packing
> >
> > Thank you for the clarification.  With the updated timeout for Bin2000, does Neural SA still outperform OR-Tools? (I would presume so, given it is superior on Bin1000.)

---

> > > ### Author Response · Authors · 2021-11-22
> > > **Response to Reviewer C2r3 (2)**
> > >
> > > We thank the reviewer for taking the time to revisit the paper and go through our rebuttal.
> > >
> > > > I agree with the authors that these works do not share the MDP formulation presented in this work (I did not, and do not, question the novelty of Neural SA itself). However my concern with the original statement, was that SA has already been combined with RL — therefore the claim that their work is “bringing it [simulating annealing] into the realm of reinforcement learning” is not correct.
> > >
> > > We agree that the wording in that sentence is not correct in that sense. We will update the text making clear that our contribution was to frame SA as an MDP and that we were not the first ones to combine SA and RL.
> > >
> > > > My comment was that the authors state that the generalisation that they present is “a truly remarkable feat, as transfer learning is notoriously difficult [in RL and CO]”, but fail to justify this very strong claim. At present, I do not feel the authors response or the promise of discussing the nuances of generalisation address this concern. Other algorithms can generalise across problem sizes/instances with vary levels of performance drop, as does Neural SA, so it is natural to ask why their performance is “a truly remarkable feat”.
> > >
> > > Thanks for clarifying this. If we may expand this discussion a bit further, it can be illustrative to compare the performance of different models when trained only on TSP20 and evaluated on TSP20, 50 and 100 (numbers taken from respective papers).
> > >
> > > |                               |    TSP20     |     TSP50     |   TSP100   |
> > > | :---                         |       :----:     |      :----:       |             ---: |
> > > | Kool et al. 2018     |   0.34%       |   ~5.0%      |   >14.0%   |
> > > | Fu et al. 2021        |   0.0000%   |   0.0145%  |   0.0370%  |
> > > | SA                         |   1.17%       |   4.34%      |   7.43%      |
> > > | Neural SA (PPO)   |   0.42%       |   1.16%      |   1.85%      |
> > > | Neural SA (ES)      |   0.10%       |   2.32%      |   5.50%      |
> > >
> > > Table: Optimality gaps for models trained on TSP20 and evaluated on TSP20, 50 and 100.
> > >
> > > If we compare our model against that of Kool et al. 2018, we see that their attention model suffers a much larger performance drop than Neural SA and even vanilla SA. Moreover, Kool et al. also observed big performance drops even when the problem size in the training set is larger than in the test set (see Figure 5 in the appendix of that paper). This shows that the data distribution is substantially distinct for different problem sizes and that machine learning approaches are very much susceptible to overfitting in this setting. In that regard, one could imagine that extending SA with learnable components could introduce similar overfitting issues, but instead Neural SA improves the generalisation ability of vanilla SA.
> > >
> > > The model proposed by Fu et al. 2021 does generalise better to larger instances, but it is worth remembering their approach requires a suite of techniques to allow a small supervised model to be applied to larger problems. These are not straightforward to implement, specific to TSP, and consist only the first step in their pipeline which still relies on a tailored Monte-Carlo tree search algorithm. In contrast, our method offers very good generalisation capabilities ‘out-of-the-box’ while using a simple, small architecture and requiring no problem-specific techniques. In this sense, we believe the generalisation capabilities of Neural SA are worth emphasising with respect to previous work.
> > >
> > > It is true that some of the generalisation ability in our method is inherited from vanilla SA, but arguably Neural SA does improve on that significantly, as shown in the results.
> > >
> > > That said, we are receptive to the reviewer’s feedback and will tone down our claims, highlighting the nuances put forward in this discussion and removing the sentence “a truly remarkable feat, as transfer learning is notoriously difficult [in RL and CO]”.
> > >
> > > > Thank you for the clarification. With the updated timeout for Bin2000, does Neural SA still outperform OR-Tools? (I would presume so, given it is superior on Bin1000.)
> > >
> > > That is right. Even when allowed to run for 2 minutes, OR-Tools does not return feasible solutions for BIN2000 other than the trivial one. So the results remain practically unchanged.

---

> > > > ### Comment · Reviewer_C2r3 · 2021-11-23
> > > > **response 2**
> > > >
> > > > I thank the authors for their constructive stance on clarifying or softening their claims to novelty and generalisation.  As stated in my original review, what stops me from providing a higher score is as follows.
> > > >
> > > > > the algorithm neither achieves SOTA performance on considered problems, or demonstrates unique benefits not found in other systems
> > > >
> > > > Whilst I am grateful for the additional data on generalisation, I do not feel it is strong enough to justify acceptance alone.  Indeed, I believe that Neural SA generalises better than, say AM, because it cannot overfit as much to the training instances.  This is because the data provided for Neural SA is that trained with PPO (Table 3), whereas if we take Neural SA trained with evolutionary strategies, the same 3 data points become 0.1%, 2.32% and 5.5% (i.e. better performing on the training set with worse generalisation).  It is undoubtably true that Neural SA generalises better than vanilla SA, however, given the weakness of vanilla SA as a baseline, this to me is not sufficient.
> > > >
> > > > Ultimately, I am still inclined to stick with my original recommendation at this time, though of course I remain open to changing this over the course of the discussion period.

---

> > > > > ### Author Response · Authors · 2021-11-25
> > > > > **Response to Reviewer C2r3 (3)**
> > > > >
> > > > > We thank the reviewer once more for the discussion and clear feedback. This has been a very insightful review, and we appreciate the reviewer taking the time to join in the discussion.
> > > > >
> > > > > > the algorithm neither achieves SOTA performance on considered problems, or demonstrates unique benefits not found in other systems.
> > > > >
> > > > > Perhaps the reviewer is too strict in his requirement of uniqueness. It is true that, for most desirable properties of CO solvers (e.g. running time, optimality gap, memory usage), Neural SA sits in between vanilla SA and other deep learning or dedicated solvers. However, we think that does not mean the trade-offs in Neural SA are not valuable for practical applications or future research. For example, when computational resources are limited or the CO problem at hand does not satisfy the assumptions of existing solvers, Neural SA can hit the right balance between solution quality, computing cost and development time. Even other SA variants require a lot of fine-tuning and would probably be harder to deploy with decent results than Neural SA.
> > > > >
> > > > > We also believe our method brings useful insights to future research, which are not immediately visible in the performance of the model. Namely, augmenting traditional and time-tested (meta-)heuristics with learnable components might be a promising direction for future research in ML4CO. In contrast to expensive end-to-end methods in previous work, this could be a more promising alternative to arrive at machine learning models capable of solving a wide range of different CO problems. Indeed, as we show in this paper, this approach can yield solid results for different problems while preserving theoretical guarantees of existing CO algorithms and requiring only simple neural architectures that can be easily trained on small problems.
> > > > >
> > > > > Also, the ICLR reviewer guide itself seems to offer more room for papers like ours: “Submissions bring value to the ICLR community when they convincingly demonstrate new, relevant, impactful, or insightful knowledge. Submissions can achieve this without achieving state-of-the-art results.” In the paper as well as in this reviewing process, we believe to have shown that Neural SA does bring new insights to the field and even might have practical value in a number of applications as discussed above, even though it does not achieve SOTA in its current form. In that sense, we do not think the paper is short of the acceptance criteria.
> > > > >
> > > > > > Whilst I am grateful for the additional data on generalisation, I do not feel it is strong enough to justify acceptance alone. Indeed, I believe that Neural SA generalises better than, say AM, because it cannot overfit as much to the training instances. This is because the data provided for Neural SA is that trained with PPO (Table 3), whereas if we take Neural SA trained with evolutionary strategies, the same 3 data points become 0.1%, 2.32% and 5.5% (i.e. better performing on the training set with worse generalisation). It is undoubtably true that Neural SA generalises better than vanilla SA, however, given the weakness of vanilla SA as a baseline, this to me is not sufficient.
> > > > >
> > > > > While vanilla SA might seem a weak baseline, we emphasise that most existent improvements for SA found in the literature are modular (Franzin & Stützle 2019) and also applicable to Neural SA, which in itself can be seen as a modular addition to the SA toolbox. Stronger results would come from searching for the best performing SA components and hyperparameters for each CO problem, which could be equally done for vanilla and Neural SA. That is certainly interesting but is perhaps outside the scope of this paper where we aimed at presenting Neural SA in its most general and simplest form.
> > > > >
> > > > > We updated the table above to include both PPO and ES results for clarity.
> > > > >
> > > > > Franzin, Alberto, and Thomas Stützle. "Revisiting simulated annealing: A component-based analysis." Computers & operations research 104 (2019): 191-206.

---

### Official Review · Reviewer_cPBg · 2021-11-03

**Correctness:** 4
**Technical Novelty And Significance:** 3
**Empirical Novelty And Significance:** 3
**Recommendation:** 6
**Confidence:** 4

**Main Review:**

Strengths:
- The paper present a simple, lightweight approach for neural simulated annealing that outperforms "vanilla SA". The approach is general and amenable to many problems.
- The approach shows promising results w.r.t generalization to larger problem sizes (a challenging aspect for many approaches to neural-guided combinatorial optimzation).

Weaknesses:
- Limited technical novelty: while the idea of learning a policy for SA using RL is novel, the technical solution follows existing approaches for policy learning using RL.
- While the experimental results are strong for a domain-independent approach, it does not seem to outperforms state-of-the-art for each problem. Further, the experiments are missing some important results. In particular:
    1) comparisons with "vanilla SA" and not state-of-the-art SA solutions that are tailored to each problem.
    2) In Knapsack, Bin packing, and TSP, the neural approaches that are tailored to the domain outperform neural SA.
    3) Results on larger sizes and additional configurations of the other neural approaches are missing, e.g., it is not clear why the results for Costa use 500 steps and the results for GAT-T use 1000 steps, or why the number of steps is fixed across larger problem sizes (while the neural SA seems to require increasingly more time to solve).

**Summary Of The Paper:**

Simulated annealing is a widely-used stochastic optimization approach. The paper presents a general approach for learning the proposal distribution for simulated annealing. The paper formalize simulated annealing as an MDP and consider the problem of learning proposal distribution as leaning a policy. They consider a lightweight policy architecture and compare two training methods, Proximal Policy Optimization (PPO) and Evolution Strategies (ES). They run experiments on four well-known benchmark domains: the Rosenbrock function, the Knapsack problem, The Bin Packing problem, and the Travelling Salesperson problem. Their results show that neural simulated annealing outperforms "vanilla SA" and that training on small problems generalizes well to larger problems.

**Summary Of The Review:**

Overall, I think it an interesting and lightweight approach that is general (amenable to many problem) and shows good performance w.r.t generalization to larger problems. However, it has limited technical novelty and the experimental results shows it is not as effective as tailored classical or neural solutions.

---

> ### Author Response · Authors · 2021-11-16
> **Response to Reviewer cPBg**
>
> We thank the reviewer for the useful feedback and comments.
>
> > Limited technical novelty: while the idea of learning a policy for SA using RL is novel, the technical solution follows existing approaches for policy learning using RL.
>
> Our paper does not introduce novel RL algorithms, but that is also the case for many influential works in machine learning for combinatorial optimisation (ML4CO) that use RL [1, 2, 4, 5]. Rather, the key insight of our paper is that a small, equivariant architecture is sufficient to parametrise a proposal distribution for simulated annealing (SA), yielding significant improvements on top of standard SA. In fact, we see it as a strength that the proposal distribution in Neural SA is easy to learn and amenable to different existing RL algorithms, which greatly simplifies the application of our method to new problems, especially for practitioners who are not RL experts.
>
> Furthermore, we believe the insight that the optimised Neural SA can get very close to many SOTA neural methods, but with just ~100 parameters, brings a new perspective to the discussion on where the community should be placing effort in ML4CO. This is namely that neural augmentation and other methods, which preserve the structure of other classical solvers can be very effective, with minimal learnable components.
>
>  > Comparisons with "vanilla SA" and not state-of-the-art SA solutions that are tailored to each problem.
>
> Most of the improvements over vanilla SA consist of adding new hyperparameters to the basic algorithm (Algorithm 1 in the paper) relating mainly to the temperature schedule but also to the acceptance criterion and exploration strategy. Latest approaches automatically search over the space of the different hyperparameters suggested in the literature to fine-tune SA to a specific problem [3]. As our goal was to study the benefits of neurally augmenting SA and not necessarily to achieve SOTA, we did not perform extensive fine-tuning either for vanilla or Neural SA. It is also worth mentioning that most of the design choices for SA found in the literature—in fact all considered in [3], except for a uniform proposal distribution—can be included into Neural SA as well.
>
> > In Knapsack, Bin packing, and TSP, the neural approaches that are tailored to the domain outperform neural SA.
>
> While that is true in many of the cases we considered, the competing architectures, e.g. [1, 5], are typically large and include many design choices that are problem specific. Conversely, Neural SA uses the same architecture for all problems, requiring little expert knowledge. This is a key detail of our model. A fair comparison would be to compare Neural SA with other "one-model-to-rule-them-all" types of networks, but we could not find any in the literature. When faced with a new problem, practitioners can readily apply Neural SA which will likely yield solid results without the need of months of research and architecture design.
>
> > Results on larger sizes and additional configurations of the other neural approaches are missing.
>
> Those results are presented as reported by the authors. All papers, including ours, use the same dataset made available in [5]. Many results for large TSP instances are missing because most of the previous deep learning methods do not scale well to problem sizes beyond 100, and the authors have not reported results on TSP200 and TSP500.
>
> > [...] it is not clear why the results for Costa use 500 steps and the results for GAT-T use 1000 steps [...] while the neural SA seems to require increasingly more time to solve.
>
> We present the results as reported by the authors who chose to evaluate their methods with a fixed number of steps. Since the size of the search space is highly dependent on the problem size, we designed our experiments with that in mind and varied the number of steps as a function of the problem size. Naturally, competing methods could yield better results with a larger number of steps, but it would be prohibitively expensive to match the number of steps used for Neural SA due to their sizeable architectures. Even with only 500 and 1000 steps, those methods are slower to evaluate than Neural SA in almost all cases.
>
> [1] Bello, I., et al. "Neural combinatorial optimization with reinforcement learning." arXiv preprint arXiv:1611.09940 (2016).
>
> [2] Deudon, M., et al. "Learning heuristics for the tsp by policy gradient." International conference on the integration of constraint programming, artificial intelligence, and operations research. Springer, Cham, 2018.
>
> [3] Franzin, A., and Stützle, T. "Revisiting simulated annealing: A component-based analysis." Computers & operations research 104 (2019): 191-206.
>
> [4] Khalil, E., et al. "Learning Combinatorial Optimization Algorithms over Graphs." Advances in Neural Information Processing Systems 30 (2017): 6348-6358.
>
> [5] Kool, W., et al. "Attention, Learn to Solve Routing Problems!." International Conference on Learning Representations. 2018.

---

> > ### Comment · Reviewer_cPBg · 2021-11-30
> > **Thank you for your response**
> >
> > I thank the authors for their detailed response!

---

### Decision · Program_Chairs · 2022-01-20

**Decision:**

Reject

**Comment:**

This work presents the Neural Simulated Annealing (NSA) approach as a heuristic for general combinatorial optimization problems. After revising the paper and reading the comments from the reviewers, here are the general comments:

- In general, the paper is clear enough. The contributions are stated in a proper way.
- The novelty is rather limited, but the key idea of using neural networks in SA, and training it with RL, has merit.
- This approach has merit but the novelty is very limited.
- The NSA improves the vanilla SA, but the benchmark reveals that NSA is not enough competitive with other state-of-the-art methods.
- The benchmark does not reveal enough information about the NSA against the SOTA methods.
- The work needs technical improvements and validation is required before accepting the work.